# Chemical zymogens for the protein cysteinome

**Mireia Casanovas Montasell** [1], **Pere Monge**[1], **Sheiliza Carmali** [1,2], **Livia Mesquita Dias Loiola**[1], **Dante Guldbrandsen Andersen**[1,3], **Kaja Borup Løvschall**[1], **Ane Bretschneider Søgaard**[1,3], **Maria Merrild Kristensen**[1], **Jean Maurice Pütz**[1] **& Alexander N. Zelikin** [1,3] ✉

We present three classes of chemical zymogens established around the protein cysteinome. In each case, the cysteine thiol group was converted into a mixed disulfide: with a small molecule, a non-degradable polymer, or with a fast-depolymerizing fuse polymer ($Z_{LA}$). The latter was a polydisulfide based on naturally occurring molecule, lipoic acid. Zymogen designs were applied to cysteine proteases and a kinase. In each case, enzymatic activity was successfully masked in full and reactivated by small molecule reducing agents. However, only $Z_{LA}$ could be reactivated by protein activators, demonstrating that the macromolecular fuse escapes the steric bulk created by the protein globule, collects activation signal in solution, and relays it to the active site of the enzyme. This afforded first-in-class chemical zymogens that are activated via protein-protein interactions. We also document zymogen exchange reactions whereby the polydisulfide is transferred between the interacting proteins via the "chain transfer" bioconjugation mechanism.

Mechanisms of reversible enzyme (de)activation in nature form the basis of signal transduction cascades and also lay behind the ability of cells and multicellular organisms to maintain the overall enzymatic homeostasis without wasteful, repetitive protein degradation and synthesis[1–3]. Nature has developed robust mechanisms to perform zymogen-enzyme interconversion, most notably via proteolytic steps, (de)acetylation, or the reversible (de)phosphorylation. In contrast, design of chemical, synthetic zymogens has proven to be challenging. First of all, it requires the identification of an essential amino acid in the protein sequence that can be modified so as to exert reversible masking of catalytic activity, and the identification of associated chemistry to achieve this[4–7]. In this regard, the most promising opportunity is associated with the protein cysteinome: modification of cysteine thiols into mixed disulfides within the active site of cysteinyl protease[8] or peptidase[4], or catalytically non-essential thiol proximal to the active site in a kinase[9] has been shown to afford reversible enzyme deactivation. Nevertheless, compared to thousands of proteins that comprise the human cysteinome, and despite the growing importance

of cysteinome inhibitors in medicinal chemistry[10], synthetic zymogens built around the cysteine thiol are solitary. Furthermore, successful in their own right, in all cases, known synthetic zymogens have been activated by external small molecule chemical stimuli and not via protein–protein interaction, as is the hallmark of biological enzyme-zymogen interconversions. This highlights the second, critical challenge in the zymogen design, namely the steric hindrance created by the protein globule, which protects the protein interior (cysteine thiols) from access by protein activators[8].

In this work, we address the fundamental challenge of developing chemical zymogens and specifically focus on the design of zymogens amenable for activation via protein–protein interactions. Towards the overall goal, we establish three types of chemical zymogens, in each case converting the protein thiol into a mixed disulfide linkage: zero-length zymogen $Z_0$ (methyl disulfide-blocked essential thiol), polyethylene glycol-based $Z_{PEG}$, and zymogens featuring a fast-depolymerizing fuse polymer based on the naturally occurring, water soluble lipoic acid, $Z_{LA}$. The latter is designed to overcome the

[1]Department of Chemistry, Aarhus University, 8000 Aarhus, Denmark. [2]School of Pharmacy, Queen's University Belfast, Belfast, UK. [3]iNano Interdisciplinary Nanoscience Centre, Aarhus Uninversity, 8000 Aarhus, Denmark. ✉e-mail: zelikin@chem.au.dk

steric constraints exerted by the protein globule, specifically so that the macromolecular fuse extends from a thiol in the protein globule into the solution bulk, wherein it interacts with the protein activators and is able to transfer such chemical information to the enzyme active site (Fig. 1a). The three designs are applied to two cysteine proteases and a kinase, and investigated in terms of masking of enzymatic activity, recovery of catalysis by small molecule activators, and zymogen reactivation by protein activators. We also investigate the zymogen exchange reactions whereby the masking group is transferred from one protein to another. Results of this study illustrate the examples of chemical zymogens reactivated using protein activators and present zymogen exchange reactions between two or more proteins.

## Results

### Polydisulfide polymer

Design of the macromolecular fuse was carried out using self-immolative polymers (SIPs)[4,11–17]. These polymers undergo triggered end-to-end decomposition when a stimulus is applied at a polymer chain end. SIPs are important for polymer recycling[18,19], drug delivery[11,20], biosensing[21,22], and materials science[23–27], and hold immense potential for diverse areas of science and (bio)technology. Of the developed SIP, polydisulfides based on the derivatives of lipoic acid (LA PDS, Fig. 1b) are unique in that these polymers are water soluble and to our knowledge are the only SIPs based on a naturally occurring compound[28–31]. Currently, the prime utility of LA PDS is in the field of intracellular drug delivery, due to the capacity of a disulfide bond to undergo degradation upon cell entry[32–35]. Herein, we propose that the "green" characteristics of LA PDS make it highly attractive for a unique application, namely engineering a macromolecular fuse to collect and propagate chemical information between two proteins, to achieve activation of chemical zymogens, and to engineer zymogen interconversion.

The application of LA PDS as a macromolecular fuse to collect and propagate a chemical signal requires an understanding of the polymer decomposition mechanism. We synthesized LA PDS via the ring-opening polymerization of the monomer, using iodoacetamide (IAm) as a chain terminating reagent. Polymer composition was confirmed via MALDI (Fig. 1c), which illustrated that the obtained polymers had lipoic acid as a monomer unit. Size-exclusion chromatography (SEC)

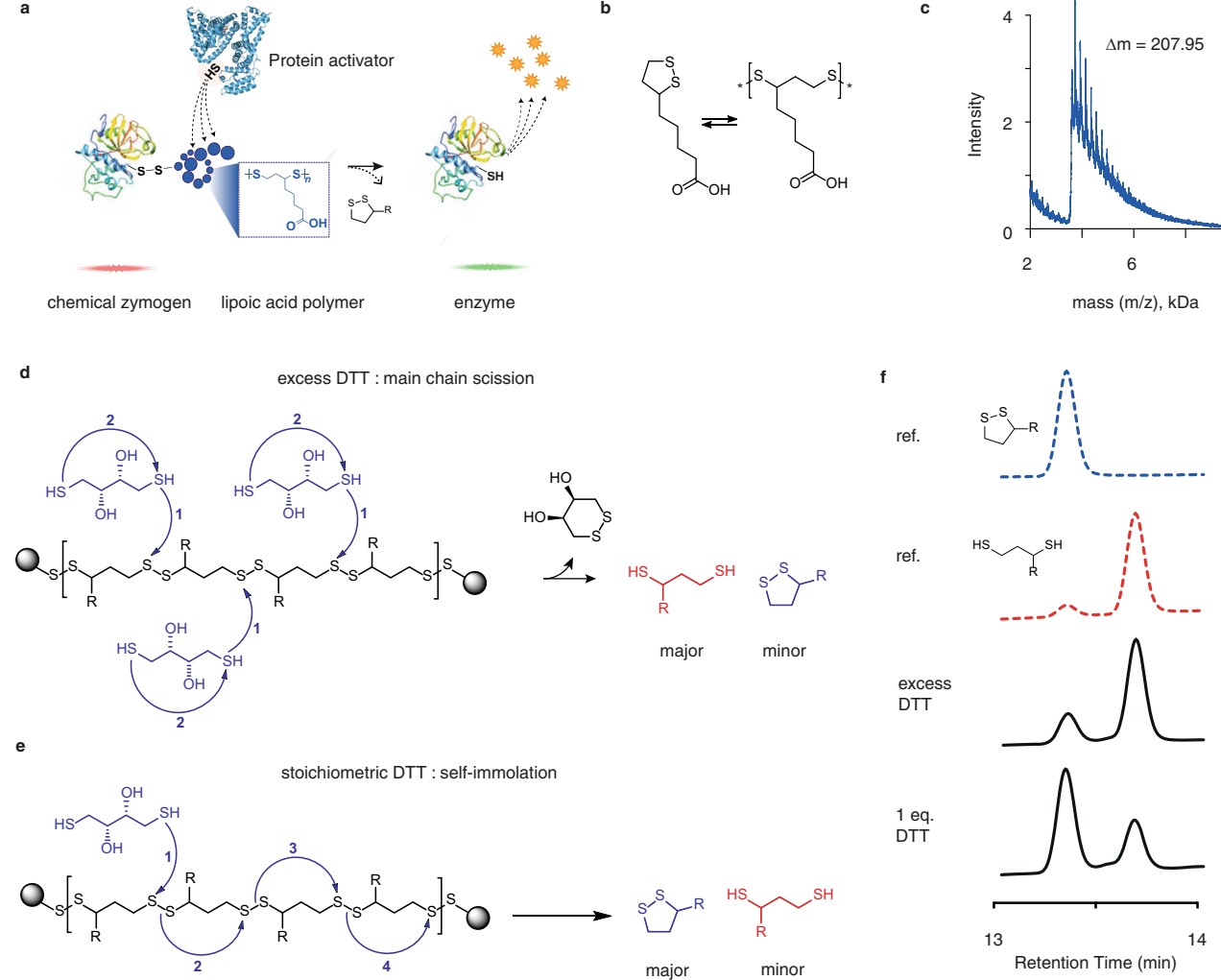

**Fig. 1 | Polydisulfide polymer based on lipoic acid. a** Schematic illustration of the proposed self-immolative polydisulfide as a tool to collect activating chemical signal in solution and propagate it to the sterically hindered active site of an enzyme; **b** Chemical formula of lipoic acid and schematic illustration of its reversible polymerization into a linear disulfide; **c** MALDI spectrum of the lipoic acid polymer that shows inter-peak spacing of 207.95 Da, in agreement with the structure of lipoic acid as the monomer unit; **d**, **e** schematic illustration of mechanisms of depolymerization for LA PDS via main-chain scission (**d**) or self-immolation (**e**), depending on stoichiometry of added reducing agent; **f** reference HPLC traces of oxidized and reduced lipoic acid monomers, and experimental data on depolymerization of LA PDS in excess DTT and with equimolar DTT, illustrating main-chain scission or self-immolation as predominant mechanisms of depolymerization, depending on stoichiometry of reaction.

revealed typical molar masses of 20–40 kDa and dispersity indexes for the synthesized polymers between 1.2 and 1.6. Triggered polymer decomposition was monitored using HPLC, which revealed that LA PDS degrades via two distinctly different mechanisms, depending on the stoichiometry of the added reducing agent (Fig. 1d–f). In excess dithiothreitol (DTT), polymer decomposition afforded *reduced* lipoic acid as the main product (Fig. 1d, f), indicative of the main-chain scission of the polymer at multiple disulfide linkages. In contrast, one mole-equivalent DTT to polymer chains afforded oxidized lipoic acid as the main product of depolymerization. This implies that added DTT exerted a single point scission of the polymer backbone, and decomposition of LA PDS proceeded predominantly via the self-immolation mechanism, that is, through sequential ring-closing of monomer units (Fig. 1e, f). Self-immolation is most desired for the envisioned use of LA PDS as a macromolecular fuse in that signal collection is propagated from the initiation point to the chain termini. Noteworthy, to our knowledge, there is only one, very recent prior example of a polymer that can degrade by two mechanisms, main-chain scission or self-immolation, with kinetics of the latter process measured in hours[36]. In contrast, depolymerization of LA PDS was rapid and complete within minutes (Fig. 1f). Moreover, initiation could occur at any disulfide bond in the backbone, and not necessarily at the chain end, as is most typical for other SIP systems[13,14].

## Zymogens design and reactivation

The zymogen design was first performed for a cysteine protease, papain. Zymogen synthesis was performed via the "grafting from" ring-opening polymerization of LA[4] with papain acting as an initiator (Fig. 2a). Resulting protein preparations exhibited blocked catalytic activity, which was readily restored via the triggered depolymerization of the macromolecular fuse (Fig. 2b). Polymerization could be terminated using a range of thiol traps such as maleimide, dithiodipyridine, sulfone, or iodoacetamide, which offers further opportunities for advanced bioconjugation endeavors (Fig. 2c). Surprisingly, non-quenched polymerization also offered a zymogen preparation, most likely via oxidative coupling/chain growth termination. In all cases, zymogens were readily reactivated into the corresponding enzyme via polymer decomposition (Fig. 2c). Noteworthy, LA PDS macro-molecular fuse was essential in the zymogen design and a direct reaction between IAm and papain afforded irreversible protein inhibition with a minimal catalytic output (Fig. 2d). Finally, zymogen reactivation was found to be a pH-dependent process, being slowest at pH 4 and rather similar at pH 6 to 8, consistent with the thiol-disulfide exchange kinetics (Fig. 2e).

To conduct a broader investigation of opportunities in the zymogen design and reactivation, two more classes of zymogens were synthesized for papain, namely the zero-length zymogen ($Z_0$) and the PEG-based counterpart ($Z_{PEG}$). In each case (as well as for the $Z_{LA}$), the essential protein thiol was modified into a mixed disulfide functionality with a methyl group for $Z_0$ or a 6 kDa polymer for $Z_{PEG}$. All zymogens were analyzed for quenching and recovery of enzymatic activity via the activity-based read-out, and for composition via MALDI (Fig. 3a). The latter analysis revealed the expected minimal change in molar mass for $Z_0$ and a change of ~5800 Da for $Z_{PEG}$, consistent with the addition of a single PEG chain to papain, as well as the broad peak for $Z_{LA}$ with masses higher than the parent protein. We note that MALDI analyses are typically accompanied with the in-flight decomposition of the labile disulfide functionality[37,38], which explains the observation of the pristine protein in the $Z_{PEG}$ preparation; it also explains only moderate molar mass increase for $Z_{LA}$. Nevertheless, for each zymogen type, enzymatic activity of the protein was masked to negligible levels, indicating quantitative modification of the essential thiol into mixed disulfides. Addition of a small molecule reducing agent (DTT) readily restored enzymatic activity, validating that each of the three zymogen designs was successful. Quantitatively similar results were obtained for

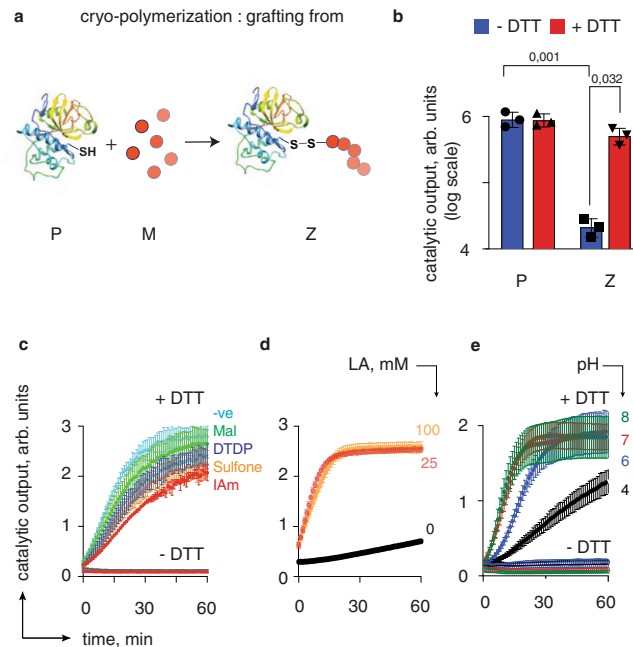

**Fig. 2 | Design, synthesis, and characterization of the polydisulfide-based zymogen of papain. a** Schematic illustration of zymogen preparation via the "grafting from" polymerization route; P protein, M monomer; Z zymogen; **b** End-point measurements of enzymatic activity for P and Z, illustrating successful deactivation of papain by LA PDS and reactivation of catalysis via polymer decomposition; **c** Kinetic data for enzymatic catalysis in solutions of zymogens obtained from papain via polymerization of lipoic acid capped with thiol traps: IAm = iodoacetamide, DTDP = 2,2′ dithiodipyridine, Mal = 4-maleimidobutyric acid; Sulfone = phenyl vinyl sulfone; −ve = non-capped cryo-polymerization); **d** Enzymatic kinetics data in solutions of papain exposed to iodoacetamide with or without polymerized LA; **e** pH dependence of kinetics of zymogen reactivation. Experimental conditions; [P, Z] = 2 μM; enzyme substrate: N-α-benzoyl-L-arginine-7-amido-4-methylcoumarin hydrochloride, 5 μM; 25 mM borate buffer pH 8 (for buffer details in panel **e**: see experimental section); DTT 2 mM. Panel **c**: shown representative results from individual runs; panels **b**, **d**, **e** shown are results from three independent experiments as mean ± S.D. Statistics in panel **b** one-way ANOVA with the Sidak's multiple comparisons test, *$p < 0.05$; ***$p < 0.001$.

another cysteinyl protease, bromelain: for each zymogen type ($Z_0$, $Z_{PEG}$, $Z_{LA}$) we achieved exhaustive suppression of enzymatic activity of the protease, which was restored in the presence of the disulfide reducing agent (Fig. 3a). This illustrates an adaptable character of the developed protocols.

To broaden the scope of the chemical zymogen methodology even further, three types of zymogens ($Z_0$, $Z_{PEG}$, $Z_{LA}$) were also synthesized for a member of the kinase protein family, creatine kinase (CK). This protein has four cysteine thiols, one of which is found in the proximity of the active site but is nevertheless non-essential for catalysis[39]. Gel electrophoresis and MALDI analyses confirmed addition of multiple PEG chains per protein for $Z_{PEG}$, and the increase in molar mass for $Z_{LA}$ over pristine kinase (Fig. 3a, b). For each zymogen composition, gel electrophoresis confirmed that the protein was released from its conjugates upon addition of DTT (Fig. 3b). By mechanism of action, CK uses creatine phosphate as a substrate and catalyzes conversion of ADP into ATP. Taking advantage of this, kinase activity was quantified through a secondary read-out, namely quantification of ATP via the luciferase-based assay. Solutions of zymogens and luciferase (supplemented with ADP, creatine phosphate, and luciferin) exhibited minor luminescence (Fig. 3a). Upon addition of DTT, solutions revealed progressively higher luminescence with time, indicative of the build-up of concentration of ATP and thus illustrating recovery of the kinase from its zymogens. Imaging of the reaction wells

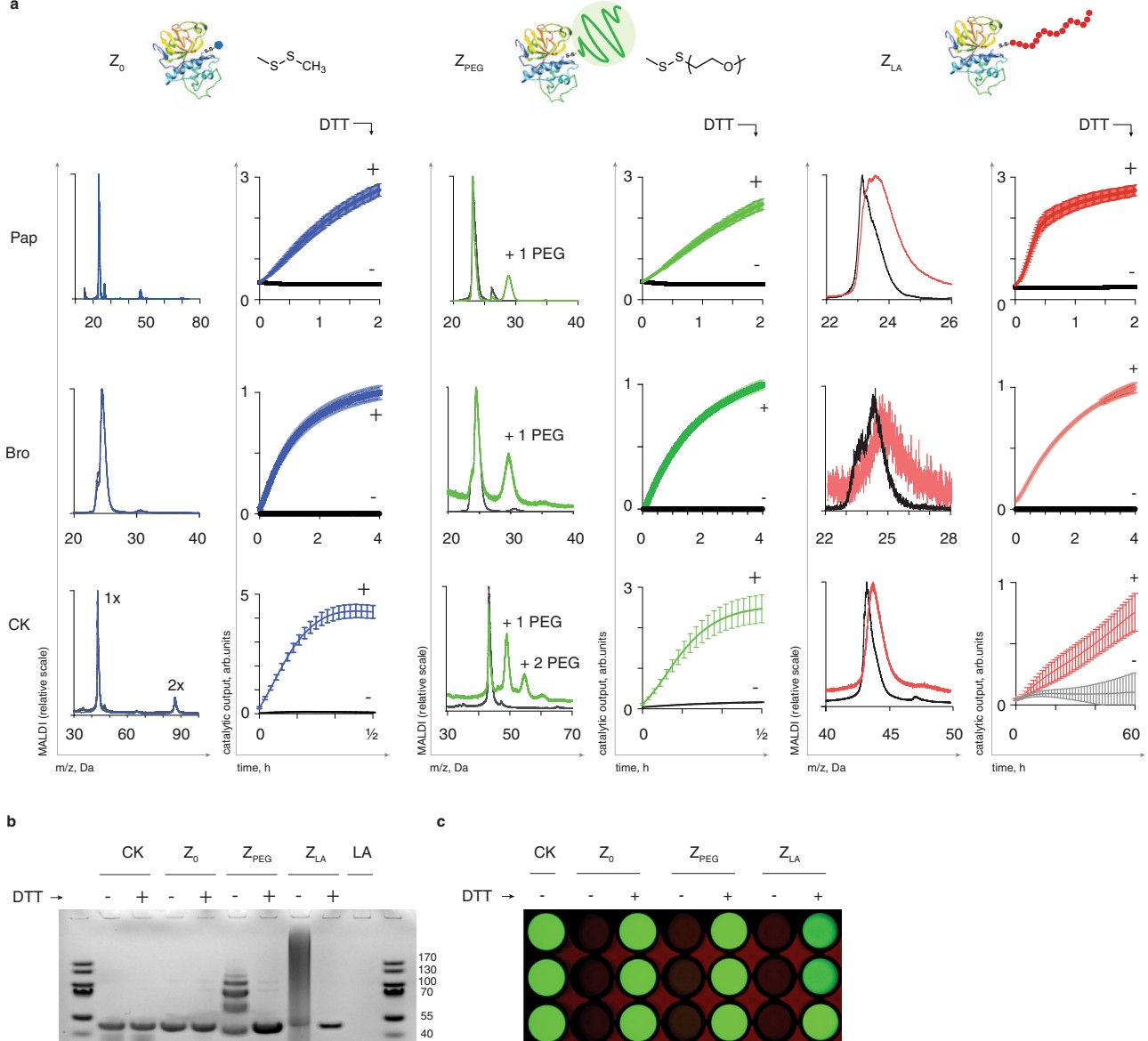

**Fig. 3 | Design and characterization of zymogens for cysteinome. a** Schematic illustration of the zymogens design, the chemical formulas of the corresponding disulfides, the reactivation data, and the MALDI characterization data for the zero-length zymogen $Z_0$, PEG-based zymogen $Z_{PEG}$, and the zymogen with the macromolecular fuse $Z_{LA}$; experimental data are shown as mean ± S.D. based on three independent experiments; **b** gel electrophoresis characterization of the creatine kinase zymogens $Z_0$, $Z_{PEG}$, and $Z_{LA}$; **c** luminescence end-point imaging of the coupled kinase/luciferase assay illustrating activity of CK masked in the composition of the three zymogens and recovered with the use of DTT (2 mM) as a reducing agent. Pap papain, Bro bromelain, CK creatine kinase.

provided visual confirmation of the zymogen reactivation for $Z_0$, $Z_{PEG}$, $Z_{LA}$ (Fig. 3c).

Taken together, results in Fig. 3 illustrate successful design of the three types of chemical zymogens, applied to three different proteins. We believe that the design methodology presented herein is easy-to-use and applicable to a diverse range of proteins that comprise cysteinome, including the proteins with thiols that are essential for catalytic activity and with thiols that are non-essential for catalysis. Our results also provide structural diversity for the zymogen design (zero length, non-degradable protein, fast-depolymerizing macromolecular fuse) to suit particular applications in biotechnology or biomedicine.

We observed a drastic difference in the zymogen reactivation phenomena between $Z_0$, $Z_{PEG}$, $Z_{LA}$ in the context of protein-mediated zymogen reactivation. To demonstrate this, we used papain-based zymogens. As protein activators, we used lysozyme, creatine kinase, pyruvate kinases (type VII and II, PK VII and II, respectively), and transglutaminase (TG). For $Z_0$ and $Z_{PEG}$, we observed negligible

catalysis on a fluorogenic papain substrate in the presence of protein activators, indicating negligible zymogen reactivation (Fig. 4a and Supplementary Fig. 1). This was also true for $Z_{LA}$ mixed with lysozyme, a thiol-free protein taken as a negative control. In stark contrast, for the thiol-containing proteins used as activators for $Z_{LA}$, we observed a significant increase in solution fluorescence, which reports on the zymogen reactivation. For CK and PK VII, this was observed at sub-micromolar $Z_{LA}$ content, whereas for PK II and TG it required a higher $Z_{LA}$ concentration of at least 3 μM. In all cases, enzymatic activity of papain recovered from its zymogen was statistically significant for $Z_{LA}$ compared to both $Z_0$ and $Z_{PEG}$ in the presence of the same protein activator.

To gain insight into the interactions between $Z_{LA}$ and the protein thiol groups on molecular level, we turned to computational analysis. First, we computed the partial atomic charges for disulfide-linked sulfur atoms within the lipoic acid polymer (modeled on a trimer). The calculated values (ca. −0.09) differed significantly from

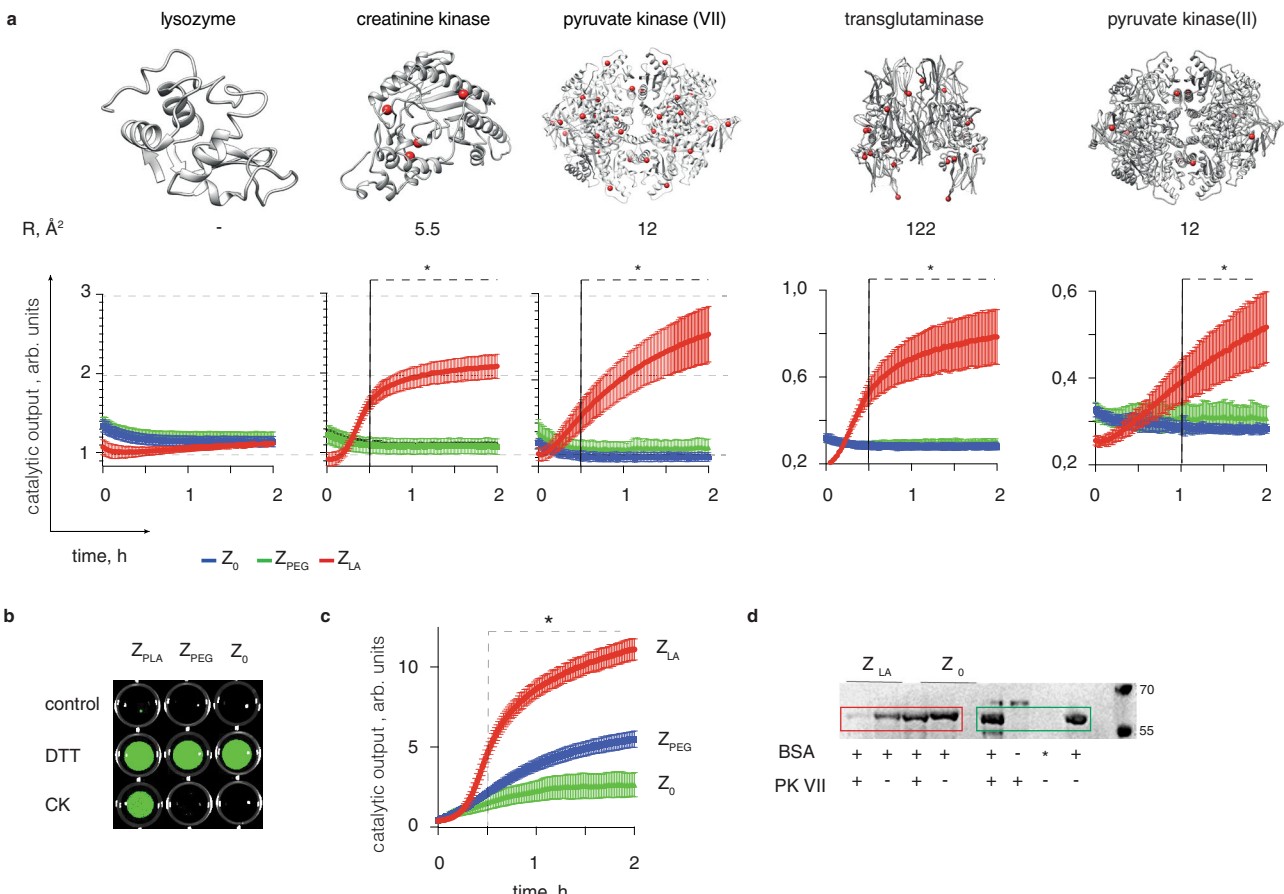

**Fig. 4 | Reactivation of zymogens using thiol-containing proteins. a** Kinetics of zymogen reactivation using protein activators (25 mM borate buffer pH 8, 25 mM NaCl, 1 mM EDTA; 10 μM papain, 1 μM protein activators); for reactivation with creatine kinase and pyruvate kinase VII, concentrations were $Z_0$, $Z_{PEG}$ = 1 μM, $Z_{LA}$ = 0.3 μM; for reactivation with transglutaminase and pyruvate kinase II: $Z_0$, $Z_{PEG}$ = 10 μM, $Z_{LA}$ = 3 μM; protein activators are shown alongside their computed surface accessibility of the thiols (R, Å$^2$). **b, c** End-point fluorescence imaging (**b**) and kinetic measurements (**c**) for reactivation of papain zymogens using creatine kinase illustrating that papain activity is masked by the three zymogens and revealed by DTT for $Z_0$, $Z_{PEG}$, and $Z_{LA}$, whereas recovery with the protein activator is possible only for $Z_{LA}$; data were recorded using albumin as a protease substrate (blocked Cys-34, labeled with fluorescein isocyanate over the level of self-quenching); In panels **a, c** data shown are an average of three independent experiments (starting with independent zymogen syntheses) shown as mean ± st.dev., statistical analysis was carried via the two-way ANOVA test with statistical significance indicating the time-point at which $Z_{LA}$ reactivation becomes statistically significant compared to both, $Z_0$ and $Z_{PEG}$, $p < 0.05$; **d** gel electrophoresis analysis of proteolytic processing of albumin (Cys-34 blocked) by papain, native or recovered from its zymogens $Z_{LA}$ and $Z_0$ by added pyruvate kinase (PK VII). In this panel, *Indicates a treatment with albumin by native papain.

those for a typical non-perturbed cysteine (ca. −0.75)[40], which indicates favorable conditions for a nucleophilic attack from the protein Cys at the polymer disulfide. Next, we calculated the surface accessibility and pKa of the cysteine thiol groups (R, Å$^2$; Fig. 4a), but observed no correlation between these values and the zymogen reactivation capability of the proteins. In fact, transglutaminase, a protein with the most surface accessible thiols, was an inferior zymogen reactivator when compared to creatine kinase, the protein with the least accessible thiol. This observation highlights that LA PDS as a macromolecular fuse effectively bypasses the steric shields and is capable of interacting even with the ill-accessible thiols in proteins. Finally, we considered that reactivity of proteins relates to the pKa of its thiol group. Indeed, Cys thiol in CK is characterized by a pKa value as much as 3 pH units lower than that for the typical Cys thiol in proteins, due to the microenvironment surrounding this cysteine[39,41], which should render it much more reactive towards thiol-disulfide exchange at pH 7−8. Nevertheless, variation in pKa values alone may not be sufficient to explain significant changes in protein thiol reactivity[42]. We believe that data in Fig. 4 illustrate that $Z_{LA}$ may serve as a tool to experimentally map reactivity of thiols in proteins, with high relevance to protein biochemistry and drug design[10].

To increase the biological relevance of the data, we realized a three-enzyme cascade, whereby creatine kinase acted as an activator to convert $Z_0$, $Z_{PEG}$, and/or $Z_{LA}$ into active papain, and activity of the latter was registered through monitoring of proteolytic degradation of albumin (blocked at Cys-34 and fluorescently labeled over the self-quenching level). This three-protein cascade was visualized using a digital camera (Fig. 4b) and quantitatively analyzed using fluorescence read-out (Fig. 4c). The three zymogens were equally powerful as zymogen design chemistries in masking papain proteolytic activity, and revealing it upon a treatment with DTT agent (Fig. 4b). Fluorescence-based activity readouts illustrated that $Z_0$ could not be activated using the kinase activator, $Z_{PEG}$ exhibited only a minor reactivation, whereas $Z_{LA}$ equipped with a macromolecular fuse was reactivated into the catalytically active papain in a statistically significant manner (Fig. 4c). These data were corroborated by gel electrophoresis analysis of albumin digestion, confirming efficient conversion of $Z_{LA}$ but not $Z_0$ into catalytically active papain by creatine kinase (Fig. 4d). Connectivity at the papain Cys thiol for each zymogen ($Z_0$, $Z_{PEG}$, $Z_{LA}$) is a disulfide linkage and zymogen constructs only differ in steric aspects. Zymogen reactivation for $Z_{LA}$ was therefore achieved specifically owing to the macromolecular fuse, which extends from the protein essential thiol into solution bulk, to interact with activators and

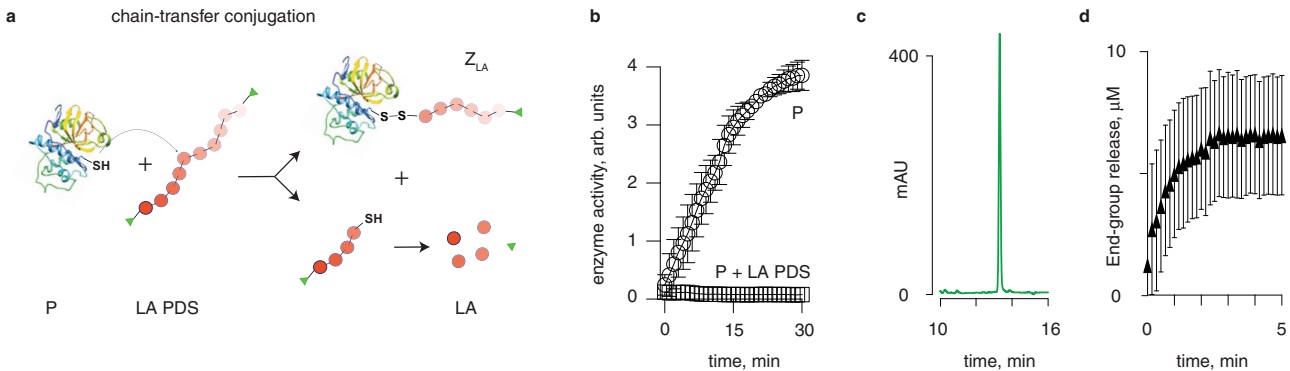

**Fig. 5 | Chain-transfer conjugation reaction between the polydisulfide polymer and papain. a** Schematic illustration of the reaction between LA PDS and a protein; **b** enzymatic activity of papain before and after reaction with LA PDS, **c** quantification of lipoic acid released during conjugation of LA PDS to papain via the chain transfer mechanism; **d** UV–Vis spectroscopy kinetics measurement of release of LA PDS thiopyridine terminal groups, resulting from a conjugation reaction between the polymer and papain. Panels **b**, **d** shows results are an average of three independent experiments, error bars signify st. dev.

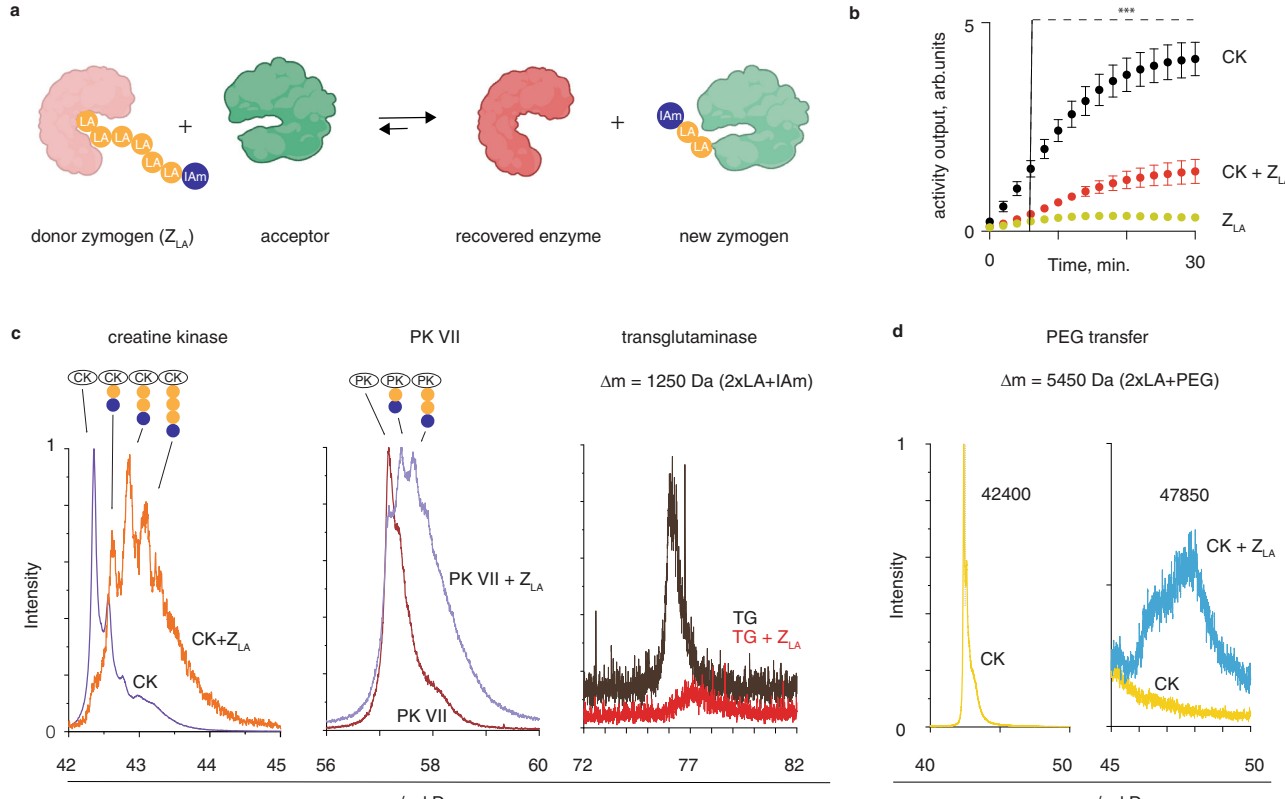

**Fig. 6 | Zymogen exchange reactions. a** Schematic illustration of a zymogen exchange reaction; **b** coupled luminescence-based read-out for activity of CK, which illustrates a loss of activity for the kinase upon an addition of papain $Z_{LA}$ as a result of the zymogen exchange reaction; shown are the results from three independent experiments as mean ± S.D; statistical analysis was carried via the two-way ANOVA test with statistical significance indicating the time-point at which the kinase activity in the presence of $Z_{LA}$ becomes statistically significant compared to both, $Z_{LA}$ and CK, $p < 0.001$; **c** MALDI-based characterization of the chain transfer reaction in solutions of papain $Z_{LA}$ and CK, PK VII, or transglutaminase; **d** MALDI data illustrating a "trans-PEGylation": a chain transfer reaction between papain $Z_{LA}$ and CK, whereby $Z_{LA}$ contained a 5 kDa PEG as a terminal group. For reaction conditions, see Experimental section.

thereafter propagate the activation signal to the enzyme active site via depolymerization. Zymogen reactivation occurred within minutes, owing to the fast depolymerization kinetics of LA PDS.

To confirm the direct reaction between a protein Cys and the polydisulfide (as occurs between a protein activator and the macro-molecular fuse within the structure of $Z_{LA}$), we used LA PDS with chromogenic thiopyridine terminal groups and papain. Thiol-disulfide exchange between the polymer and the protein should produce $Z_{LA}$ and also produce a polymer chain with an exposed terminal thiol group, which should rapidly decompose into LA monomer units and one copy of a terminal group (Fig. 5a). Indeed, the polydisulfide readily reacted with papain, as was evidenced by the complete deactivation of catalytic activity for the enzyme (Fig. 5b) accompanied by release of LA (Fig. 5c). The rate of release of the terminal groups from the structure of LA PDS was quantified via UV–Vis spectroscopy. This analysis revealed that the reaction was complete within a few minutes (initial rate of reaction exceeding 5 μM/min), illustrating fast reaction kinetics.

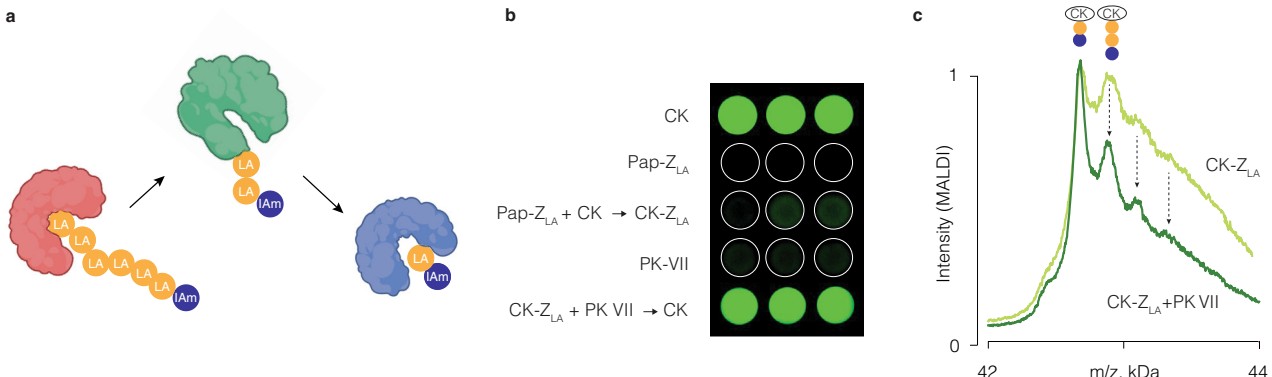

**Fig. 7 | Two rounds of transfer of the polydisulfide polymer between proteins.** **a** Schematic illustration of the two-step chain transfer; **b** visualization of luminescence (read-out for activity of creatine kinase via a coupled luciferase-based assay) in solutions of CK, papain $Z_{LA}$, and PK VII, as well as in solutions of papain $Z_{LA}$ with CK, and CK-$Z_{LA}$ with PK VII; **c** MALDI-based characterization of the two-step zymogen exchange reaction whereby CK acts first as an LA PDS acceptor and then as a polydisulfide donor. Peak intensities are normalized to that of the simplest zymogen (CK-LA-IAm).

## Zymogen exchange reactions

Results in Figs. 4 and 5 combined suggest that the overall mechanism of $Z_{LA}$ reactivation by a protein activator is a nucleophilic attack performed by the activator protein thiol onto an LA PDS disulfide. Thiol-disulfide exchange leads to the scission of the macromolecular fuse. This affords an exposed thiol on one part of the severed LA PDS, which leads to a rapid decomposition of this chain with release of LA monomer units and a terminal group, in this case—an enzyme (papain). It follows that the second part of LA PDS remains attached to the protein activator and the overall process is a "chain transfer reaction" during which a part of LA PDS is transferred from one protein to another (Fig. 6a). In other words, by activating the papain $Z_{LA}$, creatine kinase protein should be converted from an active enzyme to the corresponding zymogen, via a "zymogen exchange" reaction. To validate this, we quantified the kinase activity via a coupled luciferase read-out. To avoid proteolytic cleavage of CK and luciferase by the released papain, these experiments were conducted in the presence of a protease inhibitor. Addition of papain $Z_{LA}$ led to a pronounced, statistically significant decrease in the kinase activity (Fig. 6b), indicating conjugation of CK into the corresponding $Z_{LA}$. This result validates the occurrence of the "chain transfer" event wherein part of the LA PDS is transferred from one protein to another.

To gain quantitative information about zymogen exchange, we investigated the chain transfer processes using MALDI. We used papain $Z_{LA}$ as a donor species and a range of protein activators as LA PDS acceptors, Fig. 6c. Incubation of papain $Z_{LA}$ with creatine kinase afforded a near-quantitative conversion of the latter into its corresponding $Z_{LA}$, with the transferred linker containing the terminal iodoacetamide (IAm) group and several copies of LA monomer ($n \geq 1$, Fig. 6c). The strongest peaks observed in the MALDI spectrum correspond to transferred linkers containing between one and three lipoic acid monomer units. Quantitatively similar data were obtained for the PK VII kinase and transglutaminase, in which cases too, the transferred linker typically contained at least one or two LA monomer units. These observations suggest that the chain transfer mechanism is rather similar for all acceptor proteins: regardless of the computed protein thiol accessibility, reaction occurs at the least sterically hindered disulfides within the $Z_{LA}$, at the polydisulfide chain terminus. Nevertheless, chain transfer can occur through a thiol-disulfide exchange at a sterically hindered disulfide. Indeed, using a $Z_{LA}$ papain zymogen in which the terminal group is not IAm but a chain of PEG (5 kDa) and creatine kinase as an activator, we observed in MALDI spectrum the appearance of a peak corresponding to the transfer of the PEG chain from papain to kinase (Fig. 6d), in what appears to be an example of "trans-PEGylation" in a mixture of two proteins.

The chain transfer process based on the reversible thiol-disulfide exchange can extend over multiple rounds, using multiple proteins (Fig. 7a). To demonstrate this, we used creatine kinase first as an LA PDS acceptor and thereafter as a donor. Specifically, papain $Z_{LA}$ was mixed with CK, and thereafter this mixture was supplemented with PK VII. These reactions were conducted in the presence of a papain covalent inhibitor so that chain transfer back to papain is prevented, and also to prevent proteolytic digestion of the kinases by papain. The first round of zymogen exchange, between papain $Z_{LA}$ and CK, resulted in a transfer of LA PDS form the papain $Z_{LA}$ to CK, which is readily reported by a loss of kinase activity (Fig. 7b). Subsequent addition of PK VII (in the presence of adenosine diphosphate, creatine phosphate, luciferase, and luciferin) afforded strong luminescence. This indicates production of ATP in the solution, which in turn indicates reactivation of CK-$Z_{LA}$ by the added PK VII. Associated changes in the CK molecular weight were quantified using MALDI (Fig. 7c). Peak intensities were normalized to the intensity of the simplest zymogen, which contains only one lipoic acid monomer unit (and a IAm terminal group). Upon addition of PK VII, the intensities of peaks corresponding to CK zymogens with two or more lipoic acid units in the structure were significantly decreased (marked with arrows in Fig. 7c). This illustrates that just two lipoic acid monomer units in the structure of $Z_{LA}$ are sufficient for the chain transfer to occur. Taken together, results in Fig. 7 illustrate a transfer of LA PDS first from papain TO creatine kinase, and thereafter FROM creatine kinase to PK VII, that is, two rounds of chain transfer for LA PDS between three proteins.

Taken together, results of this study present (i) the development of chemical zymogens around the protein cysteinome, (ii) the development of chemical means to achieve zymogen reactivation using protein activators, and (iii) demonstration of zymogen exchange reactions. Three classes of zymogens were developed in this work: the zero-length $Z_0$, $Z_{PEG}$ based on a non-degradable polymer, and $Z_{LA}$ that features a fast-depolymerizing self-immolative polymer. These were applied to proteins of different classes (proteases, kinases). In each case, enzymatic activity was masked in full and reactivated by small molecule reducing agents, indicating successful zymogen design. Protein activators could only restore activity in the case of $Z_{LA}$, highlighting the critical need for the macromolecular fuse to extend into the solution bulk, collect the activation signal, and propagate it to the enzyme interior via depolymerization. We believe that significant fundamental novelty of this work also includes the "chain transfer" bioconjugation reaction between a protein and a polymer, and the observed "zymogen exchange" reaction between two enzymes. Envisioned practical applications of the developed biochemical reactions include experimentally mapping protein thiol reactivity (with

relevance to drug discovery)[43] and the use of zymogens in the design of artificial cells[44,45], which is the subject of our ongoing research.

## Methods

All chemicals and proteins, unless stated otherwise, were purchased from Sigma Aldrich and used without purification. Deuterated solvents were supplied from Euriso-Top. Ultrapure water was dispensed from MilliQ Direct 8 (Millipore) [18.2 MΩ •cm].

Nuclear magnetic resonance (NMR) spectra were recorded on a Varian Mercury 400 MHz spectrometer, running at 400 MHz. Chemical shifts ($\delta$) are reported in ppm relative to the residual solvent.

Matrix-assisted laser desorption ionization time-of-flight mass spectrometry (MALDI-TOF-MS) were recorded using a Bruker Autoflex II MS with nitrogen laser (337 nm) in linear positive 20–200 kDa mode. At least 100 laser shots covering the complete spot were accumulated for each spectrum. For molecular weight determination, sinapinic acid (20 g/L) in 50 % acetonitrile with 0.1% trifluoroacetic acid was used as matrix. Sample solution (0.1 to 1.0 g/L) was mixed with an equal volume of matrix and 4 μL of the resulting mixture was loaded onto a ground steel target plate and allowed to dry for cocrystallization.

Size-exclusion chromatography with multi-angle light scattering (SEC-MALS) was performed on a system containing an Agilent 1260 Infinity II Isocratic HPLC Pump, a Wyatt miniDAWN 3-angle static light scattering detector, an Agilent 1260 Infinity II VWD UV–Vis detector, and a Shodex RI 501 refractive index detector. The system was equipped with a Superdex 200 Increase 10/300 GL column from GE Healthcare with a length of 300 mm, an internal diameter of 10 mm, and 8.6 μm particle size. This provided an effective molecular weight range of 10,000–600,000. The eluent was 0.01 M PBS with 10% methanol and it ran at a temperature of 30 °C and a flow rate of 0.75 mL/min. Molar mass analyses were conducted using ASTRA® Software Basic.

Reverse-phase HPLC analyses were performed on an Agilent apparatus equipped with a C18 column with 2.7 μm particles, a length of 150 mm and an internal diameter of 3.0 mm from Supelco Analytical. HPLC mobile phase A was ultrapure $H_2O$ supplemented with 0.1% TFA (v/v) and mobile phase B acetonitrile supplemented with 0.1% TFA (v/v). Elution was performed starting with solvent B 5% to B 100% over 15 min, hold B 100% for 4 min at $T = 40$ °C and a flow rate of 0.4 mL/min. Detection was performed by UV detector (220 nm and 280 nm).

Fluorescence and luminescence measurements were recorded on an Enspire 2300 Multilabel Reader (Perkin Elmer®) or a SynergyH1 microplate reader (BioTek®)

### Preparation of compounds

**Solution-based ring-opening polymerization (ROP) of lipoic acid.** In a typical procedure, α-lipoic acid (300 mg, 1.5 mmol, 1.0 equiv.) and 2-iodoacetamide (5.4 mg, 29.1 μmol, 0.02 equiv.) was added to a 5 mL pre-dried pear-shaped flask and dissolved in 100 μL dry DMF, heated to 70 °C and stirred magnetically under and argon atmosphere for 2 h to yield a dark-brown solution. Hereafter, NaHCO₃ (pH 8.5, 0.4 M) was added to the reaction mixture to deprotonate the carboxylic acid thereby solubilizing the polymer. A spatula was used to release the stirring bar from the glass flask and the reaction mixture was heated to 40 °C to speed the process. The resulting solution was transferred to a dialysis membrane (MWCO 3.5 kDa) and dialyzed against water. The polymer was recovered via freeze-drying as a white fluffy solid (166 mg, 55%). ¹H-NMR (400 MHz, D₂O), $\delta$, ppm: 3.29 (s, 4H), 2.98–2.85 (m, 59H), 2.36–1.98 (m, 84H), 1.77–1.36 (m, 122H). Mn (NMR) = 4.2 kDa, Mn (SEC-MALS) = 9.9 kDa, Đ = 1.3

**Reduction of papain and bromelain.** Papain and bromelain, as cysteine proteases, are commercially provided as partially inactivated enzymes. Activation is accomplished by incubation with a thiol-containing compound or a reducing agent. In a typical experiment, to a solution of papain (5 g/L, 0.17 mM, 0.5 mL) in 50 mM phosphate buffer pH 6.8 with 10 mM EDTA was added tris (2-carboxyethyl) phosphine hydrochloride (TCEP, 1.1 mM, 5 equiv). The reaction was incubated at 37 °C for 1 h. After this time, TCEP was removed by gel filtration (NAP-5, GE Healthcare) in the same buffer and concentrated using spin filtration (MWCO 3 kDa).

**Preparation of PEG-S-S-TP.** A solution was prepared of 2,2′-dithiodipyridine (45.8 mg, 4.0 equiv.) and acetic acid (1.5 μL, 0.5 equiv.) in methanol (7.5 mL) under an atmosphere of argon. MeO-PEG-SH-6000 (312.0 mg, 1.0 equiv.) was dissolved in a mixture of methanol:acetic buffer 50 mM (pH = 4.5) 1:1 v:v (2 mL), purged with argon and added dropwise to the reaction mixture. The reaction was left stirring at room temperature for 21 h. The product was precipitated twice into cold diethyl ether followed by centrifugation. The precipitated product was obtained by filtration followed by solvent removal in vacuo to yield PEG-S-S-TP (176.0 mg, 53%) as a white solid. ¹H-NMR (400 MHz, Methylene Chloride-$d_2$) $\delta$ 8.45–8.41 (m, 1H), 7.83–7.78 (m, 1H), 7.71–7.65 (m, 1H), 7.13–7.07 (m, 1H), 3.77 (t, $J = 5.2$ Hz, 2H), 3.71–3.41 (m, 144H), 3.34 (s, 3H), 3.00 (t, $J = 6.0$ Hz, 2H).

**General protocol for zero-length zymogens.** Reduced papain, bromelain (2 g/L) or creatine kinase (1 g/L) were reacted with S-methyl methanethiosulfonate (MMTS, 500 equiv.) in sodium phosphate buffer (50 mM, 10 mM EDTA, pH 6.8). The solution was stirred at 4 °C overnight and the proteins were purified/buffer exchanged to sodium acetate buffer (2.5 mM, pH 4.5, 50 mM NaCl) using Amicon centrifugal filters (MWCO 3 kDa).

**General protocol for PEG-based zymogens.** Reduced papain and bromelain (100 μM) or creatine kinase (25 μM) were combined with 10 eq. of PEG-TP (synthetized as described above or commercially available MeO-PEG-OPSS 5 kDa from Iris Biotech) in acetate buffer (2.5 mM, pH 4.5, 50 mM NaCl) and incubated at 37 °C for 1 h. After this time, the crude sample was purified by gel filtration (NAP-5) in the same buffer.

**General "chain transfer" protocol for preparation of LA PDS zymogens.** Reduced papain or bromelain (2 mg/mL) and excess LA PDS (10×, by weight) were dissolved in sodium phosphate buffer (50 mM, 10 mM EDTA, pH 6.8). The solution was stirred at 37 °C for 2 h. After this time, the crude reactions were purified by Amicon centrifugal filtration in the same buffer.

**General "grafting from" polymerization protocol for preparation of LA PDS zymogens.** Reduced papain, bromelain (0.5 g/L) or creatine kinase (1 g/L) were dissolved in solutions containing 100 or 25 mM of LA in 0.2 M NaHCO₃, respectively, and 1 mL of these reaction solutions was kept in the freezer at −20 °C for 2 h. After this time, 200 μL of a 0.1 M solution of iodoacetamide in Milli Q water was added to the frozen reactions and left stirring at room temperature for 30 min. The zymogens were purified by gel filtration (CentriPure P10) in milli Q water.

**Preparation of fluorescently labeled, self-quenched albumin.** BSA and potassium carbonate were dissolved in milli Q water to 10 g/L each. Then fluorescein isothiocyanate (FITC) was added to the solution from a concentrated DMSO stock to a final concentration of 2 g/L (final DMSO 2% v/v). The reaction was left at 37 °C under stirring for 21 h and purified extensively by Amicon centrifugal filtration (MWCO 3 kDa) in Milli Q water until no absorbance from fluorescein could be observed in the wash waters.

**Preparation of Cys-blocked BSA.** BSA was dissolved in borate buffer (25 mM, pH 8.0) to a final concentration of 2 g/L. To that solution 350

equivalents of iodoactamide were added and the mixture was incubated at 37 °C for 30 min. After that time, the sample was purified by gel filtration (CentriPure P10) in Milli Q water.

## Experimental protocols

**RP-HPLC analysis of poly (lipoic acid) degradation using dithiothreitol (DTT).** Poly (lipoic acid) was incubated at room temperature in the presence and absence of dithiothreitol in 50 mM phosphate buffer, pH 8.0. At specific time points ($t = 5$ min and 24 h), reactions were analyzed by RP-HPLC. Lipoic acid (0.10 g/L, 0.5 µmol) incubated under the same conditions served as control.

**Reactivation of papain LA PDS zymogens.** Solutions of 1 µM of papain and papain LA PDS zymogens prepared by either chain transfer or polymerization methods were incubated with 10 µM Nα-benzoyl-L-arginine-7-amido-4-methylcoumarin (AMC) in phosphate buffer (50 mM, 10 mM EDTA, pH 6.8). To each sample was added DTT to 10 mM or just buffer as a control with final reaction volumes of 100 µL. After 60 min at 37 °C substrate hydrolysis was measured by recording fluorescence at $\lambda_{ex}/\lambda_{em}$ 370/460 nm.

**Effect of different polymerization quenchers on LA PDS zymogen reactivation.** LA PDS zymogens were prepared by polymerization as described above, but changing the quenching reagent used after polymerization. The quenchers used were: iodoacetamide, 2,2'dithiodipyridine, 4-maleimidobutyric acid and phenyl vinyl sulfone in concentrations of 0.1 M. A reaction in the absence of quencher was also prepared. After purification, equal volumes of each reaction were diluted in borate buffer (25 mM, pH 8.0) to a final protein concentration of 2 µM, combined with 5 µM of AMC. Catalytic activity in the presence or absence of DTT (2 mM) was analyzed by substrate hydrolysis in 100 µL reaction volumes as detected by fluorescence increase at $\lambda_{ex}/\lambda_{em}$ 370/460 nm at 1 min intervals for 1 h at 37 °C in a plate reader.

**Effect of LA concentration during polymerization.** LA PDS zymogens of papain were prepared using the "grafting from" polymerization protocol described above with concentrations of LA during the reaction set to 100 or 25 mM and also a control without LA. After quenching of the reactions with iodoacetamide and purification, equal volumes of each reaction were diluted in borate buffer (25 mM, pH 8.0) to 0.3 µM protein (as determined by UV absorbance at 280 nm), combined with 10 µM of AMC. Catalytic activity in the presence and absence of DTT (2 mM) was analyzed by substrate hydrolysis in 100 µL reaction volumes as detected by fluorescence increase at $\lambda_{ex}/\lambda_{em}$ 370/460 nm at 1 min intervals for 2 h at 37 °C in a plate reader. These experiments were performed in three independent replicates (independent zymogen syntheses).

**Effect of pH on the reactivation of papain LA PDS zymogens.** Papain LA PDS zymogen prepared by polymerization method as in section 2.7 was dissolved at 5 µM and combined with 50 µM AMC in different buffers with the addition of 10 mM DTT. Samples wihout DTT were prepared as controls. The different buffers were: formic acid (20 mM, pH 4.0), 2-(N-morpholino)ethanesulfonic acid (MOPS, 20 mM, pH 6.0), phosphate buffer (20 mM, pH 7.0) and borate buffer (25 mM, pH 8.0). Substrate hydrolysis was measured by recording fluorescence at $\lambda_{ex}/\lambda_{em}$ 370/460 nm at 1 min intervals for 2 h at 37 °C in a plate reader. These experiments were performed in 3 independent replicates (independent zymogen syntheses).

**Reactivation studies of papain and bromelain zymogens.** Reactivation of different zymogens ($Z_0$, $Z_{PEG}$, and $Z_{LA}$) of papain and bromelain was studied by monitoring fluorescence increase upon hydrolysis of AMC. Briefly, solutions containing papain zymogens ($Z_0$, $Z_{PEG} = 1$ µM,

$Z_{LA} = 0.3$ µM as determined by UV absorbance at 280 nm) or bromelain zymogens ($Z_0$, $Z_{PEG}$, $Z_{LA} = 5$ µM, as determined by UV absorbance at 280 nm) were combined with 10 µM of substrate and DTT (2 mM) in borate buffer (25 mM, pH 8.0) in a volume of 100 µL. Samples without DTT were used as controls. Fluorescence ($\lambda_{ex}/\lambda_{em}$ 370/460 nm) was monitored at 1 min intervals at 37 °C in a plate reader. These experiments were performed in three independent replicates (independent zymogen syntheses).

**Reactivation studies of creatine kinase zymogens.** Reactivation of different zymogens ($Z_0$, $Z_{PEG}$, and $Z_{LA}$) of creatine kinase (CK) was quantified via a coupled bi-enzymatic assay. Briefly, unmodified CK or zymogens were diluted to 100 nM (as estimated by a protein assay with fluorescamine using CK as reference) in Tris-HCl buffer (20 mM, pH 8.0) in a white microplate. Then, DTT was added, and quickly after, a solution containing ADP, creatine phosphate (CP), luciferin and Cell-Titer Glo® 2.0 Cell Viability Assay luciferase (Promega) was added to the wells. Luminescence was monitored at 37 °C in regular time intervals over 1 h (RLU integrated every 2 s) in a plate reader. Final volume in every well was 100 µL and concentration of reagents was as follows: DTT 20 µM, ADP, CP and luciferin 1 µM, Cell-Titer Glo® 2.0 1:50 dilution (v/v). Controls included samples of each zymogen/protein without DTT, CK without CP and samples without CK or zymogen. At the endpoint, luminescence was also imaged inside an ImageQuant™ LAS 4000 camera system (GE Healthcare) with 3 s exposure. These experiments were performed in three independent replicates (independent zymogen syntheses).

**Gel electrophoresis of creatine kinase zymogens.** Samples of 2 µg of creatine kinase and creatine kinase zymogens were prepared in Milli Q water supplemented with LDS buffer (Pierce, 1:4 final dilution) and NuPAGE™ sample reducing agent (in a final dilution of 1:8). For each protein or zymogen, a sample without reducing agent was also prepared. Samples were incubated 20 min at 37 °C and subsequently 10 µL were loaded into a NuPAGE™ gel (4 to 12% bis-tris, 1.0 mm) and run for 50 min in MOPS buffer at 150 V and later stained with coomassie blue.

**Reactivation of papain zymogens with different protein activators.** Reactivation of different papain zymogens ($Z_0$, $Z_{PEG}$, and $Z_{LA}$) was studied by monitoring fluorescence increase upon hydrolysis of AMC. Briefly, solutions containing papain zymogens ($Z_0$, $Z_{PEG} = 1$ µM, $Z_{LA} = 0.3$ µM as determined by UV absorbance at 280 nm) were combined with 10 µM of substrate and creatine kinase (CK) or pyruvate kinase VII at 1 µM in borate buffer (25 mM, pH 8.0). A negative control with 1 µM lysozyme (non-thiol-containing protein) was prepared in the same conditions as well. Reactivation was also performed with transglutaminase and pyruvate kinase II at 1 µM, but with a 10-fold increase in zymogen content ($Z_0$, $Z_{PEG} = 10$ µM, $Z_{LA} = 3$ µM). Fluorescence ($\lambda_{ex}/\lambda_{em}$ 370/460 nm) was monitored at 1 min intervals at 37 °C in a plate reader. All well volumes were set to 100 µL and samples with DTT (2 mM) or without any activator were prepared as positive and negative controls, respectively. These experiments were performed in three independent replicates (independent zymogen syntheses).

**Reactivation of papain zymogens by protein activator with degradation of a protein substrate.** In a 96-well plate, solution of self-quenched BSA-FITC in borate buffer (25 mM, pH 8.0) was combined with different papain zymogens ($Z_0$, $Z_{PEG} = 1$ µM, $Z_{LA} = 0.3$ µM as determined by UV absorbance at 280 nm) and 1 µM of creatine kinase in the same buffer. Reaction volume was 100 µL; samples with DTT (2 mM) and no activator were used as positive and negative control, respectively. Progress of the reaction was observed by the increase in fluorescence $\lambda_{ex}/\lambda_{em}$ 490/520 at 37 °C in a plate reader for 2 h in intervals of 1 min. After 2 h the plate was imaged inside an Image-Quant™ LAS 4000 camera system (GE Healthcare) using the GFP filter

in fluorescence mode. These experiments were performed in three independent replicates (independent zymogen syntheses).

**Reactivation of papain zymogens via protein–protein interaction imaged using SDS PAGE.** Samples with 4 µM of Cys-blocked BSA and 1 equivalent of different papain zymogens ($Z_O$, $Z_{LA}$) or pristine papain were prepared in Milli Q water and supplemented with SDS buffer (Pierce, 1:4 final dilution). For each protein or zymogen, a sample with 3.3 µM PKVII was also prepared. As positive controls, one sample containing only 4 µM of quenched BSA, and another with 3.3 µM PKVII were prepared. Samples were incubated 10 min at room temperature and subsequently 10 µL were loaded into a NuPAGE™ gel (4 to 12% bis-tris, 1.0 mm) and run for 55 min in MOPS buffer at 150 V and later stained with coomassie blue.

**Rate of chain transfer reaction.** Rate of chain transfer reaction was studied by monitoring release of the polymer end-cap, in this case thiopyridine, upon triggered depolymerization by papain. In a 96-well plate, a solution of 1.2 g/L LA PDS end-capped with thiopyridine was combined with 6 µM papain in borate buffer (25 mM, pH 8.0). Absorbance at 343 nm was monitored at 10 s intervals at 37 °C in a plate reader, and later translated to thiopyiridine concentration by using a calibration curve. All well volumes were set to 100 µL and samples with DTT (5 mM) or with the polymer alone were prepared as positive and negative controls, respectively. These experiments were performed in three independent replicates.

**Activity read-out for zymogen exchange experiment with CK.** Samples containing 10 µM (determined by UV absorbance at 280 nm) of papain zymogen ($Z_{LA}$), 27 µM papain inhibitor (E-64, Sigma), and 4 µM creatine kinase were prepared. First, papain zymogen was incubated with the inhibitor at room temperature for 30 min, and then CK was added and the mixture was incubated for 1 h more at 37 °C. After this time, samples were dissolved in Tris-HCl buffer (20 mM, pH 8.0) in a white 96-well plate to a concentration of 0.1 µM of CK. Samples with 0.1 µM CK and 0.3 µM papain zymogen were prepared as positive and negative controls, respectively. Immediately after, CK substrates ADP, CP and luciferin at a concentration of 1 µM and Cell-Titer Glo® reagent in a 1:10 dilution were added (final volume in well 100 µL) and luminescence was monitored at 37 °C in regular time intervals 1 h (RLU integrated every 2 s) in a plate reader. These experiments were performed in three independent replicates (independent zymogen syntheses).

**Zymogen exchange experiment with different proteins.** Samples with papain-LA PDS zymogen at a concentration of 8.3 µM (determined by UV absorbance at 280 nm) were incubated with 2 equivalents of papain inhibitor (E-64, Sigma) at room temperature for 30 min. Then, 3.5 µM of different protein activators (creatine kinase, transglutaminase and pyruvate kinase VII) were added and incubated for 1 h more at 37 °C. After this time samples were analyzed in MALDI-TOF-MS, together with the corresponding pristine proteins diluted in Milli Q (papain, creatine kinase, transglutaminase and pyruvate kinase VII) as controls.

**Trans-PEGylation from papain $Z_{PEG}$ to creatine kinase.** Sample containing 72 µM of papain $Z_{PEG}$ (determined by UV absorbance at 280 nm) was incubated with 2 equivalents of papain inhibitor (E-64, Sigma) at room temperature for 30 min. Then, 5 µM of creatine kinase was added and incubated for 1 h more at 37 °C. Controls with pristine papain or creatine kinase diluted in Milli Q were prepared as well. All samples were analyzed in MALDI-TOF-MS.

**Zymogen exchange experiment with two protein acceptors.** Samples with papain-LA PDS zymogen at a concentration of 8.3 µM

(determined by UV absorbance at 280 nm) were incubated with 2 equivalents of papain inhibitor (E-64, Sigma) at room temperature for 30 min. Then, 3.5 µM of creatine kinase or pyruvate kinase VII were added and incubated overnight. The next day, 1 µM of pyruvate kinase or creatine kinase, respectively, were added. Samples without the second protein addition and controls with pristine creatine kinase and pyruvate kinase diluted in Milli Q were prepared as well. Al samples were analyzed in MALDI-TOF-MS.

**Imaging the two-step zymogen exchange reactions.** Samples with papain-LA PDS zymogen at a concentration of 15 µM (determined by UV absorbance at 280 nm) were incubated with 3.5 equivalents of papain inhibitor (E-64, Sigma) at room temperature for 30 min. Then, 15 µM of creatine kinase were added and incubated for 1 h at 37 °C. After that time, the reaction was quenched with 20 mM iodoacetamide for 30' and then purified by gel filtration (CentriPure P10) in borate buffer (25 mM, pH 8.0). This sample was then transferred to a 96-well plate (per triplicate), with and without the addition of 5 µM of pyruvate kinase and incubated for 1 h more (samples with the second protein addition were first mixed with 3.5 equivalents of papain inhibitor E-64). Controls with pristine creatine kinase, papain-LA PDS zymogen and 5 µM pyruvate kinase diluted in borate buffer were prepared as well. Finally, CK substrates ADP, CP and luciferin at a concentration of 1 µM and Cell-Titer Glo® reagent in a 1:10 dilution were added and, after 40 min reaction, luminescence was imaged inside an ImageQuant™ LAS 4000 camera system (GE Healthcare) with 8 s exposure.

### Analyses

**Statistical analysis.** Where reported, statistical significance was evaluated with a two-way ANOVA with the Sidak's multiple comparisons test performed in the software Graphpad Prism®.

**Cysteine surface accessibility analysis.** The tertiary structures of glutathione reductase, pyruvate kinase, transglutaminase, protein kinase A and polynucleotide kinase were downloaded from the Protein Data Bank (PDB codes: 2HQM, 3N25, 1KV3, 4WB5, 1AQF, and 1LTQ, respectively). Molecular graphics and solvent accessible surface areas (Å²) calculations of cysteine residues were performed with UCSF Chimera (Resource for Biocomputing, Visualization and Informatics at the University of California, San Francisco, with support from NIH P41-GM103311). Accessible surface area values were determined with a probe radius of 1.4 Å, equivalent to the radius of a water molecule. As a reference for cysteine accessibility, calculated values were compared to reported accessible surface area of a cysteine residue in a folded protein (5 Å²)[46]. In this analysis, all cysteine residues with calculated values above 5 Å² were considered accessible.

**Prediction of cysteine pKa and estimation of partial atomic charges in lipoic acid.** The pKa of cysteinyl thiol groups in protein activators was predicted using PROPKA. The PROPKA method consists of an empirical approach to determining pKa values by calculating the effect of the protein environment on an amino acid side chain[47,48]. For the thiol group in cysteine residues, the starting pKa in the calculations is 9.0. Poisson-Boltzmann electrostatic surface map for a lipoic acid trimer was generated with Maestro (Release 2022-1: Maestro v.13.1.137, Schrödinger, LLC, New York, NY, 2021). The trimer of lipoic acid provided a closer representation to poly (lipoic acid) without extensive molecular dynamic simulations. Partial atomic charges for the thiols in lipoic acid were determined using Discovery Studio (BIOVIA, Dassault Systèmes, Discovery Studio Visualizer, v21.1.0.20298, San Diego: Dassault Systèmes, 2020) for comparison to cysteinyl thiol pKa values.

### Reporting summary
Further information on research design is available in the Nature Research Reporting Summary linked to this article.

## Data availability

The data that support the findings of this study are available from the corresponding author upon request. Source data are provided with this paper.

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

## Acknowledgements

Authors acknowledge financial support from the Independent Research Fund Denmark (DFF FNU Grant No. 0135-00162B, to A.N.Z.), the Novo Nordisk Foundation (grant No. NNF20OC0062131, to A.N.Z.), the Carlsberg Foundation (Grant No. CF19-0275, to A.N.Z.), and the Lundbeck Foundation (grant No. R287-2018-1117, to S.C.).

## Author contributions

Project ideation, experiment design: S.C., P.M., M.C.M., A.N.Z.; collection of experimental results: M.C.M., P.M., S.C., L.M.D.L., D.G.A., K.B.L., A.B.S., M.M.K., J.M.P.; data analysis and interpretation: M.C.M., P.M., S.C., A.B.S., A.N.Z.; manuscript writing and revision: M.C.M., S.C., A.N.Z.

## Competing interests

The authors declare no competing interests.
