## [Peer Review File · Nature Communications]

REVIEWER COMMENTS

Reviewer #1 (Remarks to the Author):

This manuscript developed the chemical zymogen (inactive proteins) based on modification of the cysteine moiety of the active protein with sulfhydryl polymers. In the presence of DTT, the polymer chains were degraded into small monomers by a "self-elimination" process, resulting in the full reactivation of the zymogen. In the preparation of zymogen, the polymer was connected to the protein surface via a chain transfer-like pathway. This "chain transfer" pathway was further used to explain the protein-protein activation process. The chains on the zymogen surface were partially transferred to the activator surfaces, and consequently, the spatial site-blocking effect of the polymer chains that exist in the zymogen disappeared with the recovery of protein activity. Conversely, the increased chain on the surface of the activator led to the inhibition of its own activity to achieve zymogen exchange. Despite the interesting finding of this phenomenon, the experimental analysis is insufficient and lacks in-depth exploration, which contributes weakly to the concepts proposed in the article. The followings are the specific concerns.

1. The authors demonstrated the concept of zymogen activation by kinetic measurements and point luminescence imaging. The chain transfer occurs between the activated protein and the zymogen, which is essentially an exchange of sulfhydryl groups, thus causing a transition between protein activities. For this reaction, results in the article demonstrate only the existence of this phenomenon without providing any quantitative data. For example, what is the molecular weight of the chain required for the reaction to occur; what is the length change of the chain after activation occurs; whether or not various proteins are different for chain lengths and numbers; whether or not a generalized reaction and modification condition can be defined, etc. Also, the rate and kinetics of the chain transfer reaction should be systemically explored.

2. The authors should provide one or more useful applications for the discovered "zymogen exchange" phenomenon. Also, the specificity of the zymogen activation in the real biological environment is questionable, because sulfhydryl groups widely exist in various bio-species. Therefore, under the complicated conditions of the biological environment, it is necessary to perform independent verification to illustrate its feasibility and significance.

3. The authors also proposed that the polymer LA PDS undergoes "chain transfer" and "self-elimination". Therefore, is it possible that the activated protein can still maintain biological activity when its concentration is sufficiently high or when it possesses exceptionally high amount of sulfhydryl groups, thus achieving real protein "green activation".

4. The authors seem not to highlight the advantages of protein modification method in their research. In my opinion, the modification process needs to be highlighted as easy-to-use, strong universality, and high stability and efficiency. For highlighting its unique role, it can be compared with common protein modification methods.

5. In order to prove the authors' conjecture, it is suggested that the interaction between polymer chains and protein sulfhydryl groups in water be experimentally verified by other theoretical calculations, such as computer simulations to further simulate the structure and "chain transfer" behavior of the polymer.

Reviewer #2 (Remarks to the Author):

The paper 'Green self-immolative polymer: molecular antenna to collect and propagate the signal for zymogen activation' investigates the development of chemical zymogens using a series of disulphide coupled additives. Most notably, they use a self-immolative lipoic acid polymer connected through a disulphide bridge. This polymer can undergo depolymerization with a protein activator, thus activating the enzyme. This is a significant advance on previous work as they demonstrate a protein can be used to generate the switch in enzyme activity and thus the science is extremely interesting and topical for a broad audience. However, I had some issues with the clarity of the data presented and thus I believe more detail is needed before their results can be properly assessed. I have highlighted these points below. I believe novelty is high and thus if these points can be addressed it will make this manuscript acceptable for publication in Nature Communications.

Figure 2B – This data is not clear evidence of the conjugation of the lipoic acid polymer. We would typically see a shift in the SEC trace upon significant increase in molecular weight (like Figure 3C). The one trace shown is in black and it is not explained in the figure, the lines in red over the top (which are supposed to correspond to the starting materials and conjugate) are firstly not SEC traces and seem on top of each other to me. Please explain in the text how this shows conjugation? They are also hard to tell apart as the colours are very similar.

Figure 2C&D – The addition of GSH/DTT concentration would add some clarity

Figure 2F – This data showed that the activity was not completely blocked for Z, without DTT. This should be discussed briefly in the text.

Figure 3 – The first two enzymes have similar activity when cleaved regardless of the additional moiety. In contrast, creatine kinase has completely different activity for each moiety over the time frame of the experiment, why is this? Does this mean there are differences in cleavage rates?

Figure 4B – The authors state that only the ZLA can be activated by the protein. However, figure 4B appears to show that there is increase in activity for ZPEG as well (approx. half), this doesn't seem to correlate to Figure 4C where I would expect to see a low fluorescence. Can the authors explain this?

Figure 4 – The authors use pH 8, 25 mM Borate buffer, why is this? It doesn't look like pH is an issue but does high salt effect zymogen switching? I would like to see one experiment done in standard physiological conditions (phosphate buffer pH 7.4) to assess if this approach is useful under these conditions.

Reviewer #3 Comments

Remarks to the Author:

The authors prepared three types of zymogens through disulfide linkages including ZO, ZPEG and ZLA. The design strategy was applied to two thiol-containing proteases and a kinase. As a result, the bioactivities of the enzymes were blocked and could be restored by reducing small molecules. Among the zymogens, only ZLA modified by poly(lipoic acid) could be reactivated by protein activators. They also demonstrate a unique zymogen exchange reaction between two proteins. The results and methods reported here have the potential to facilitate the development of new chemical zymogens. However, the data provided here are not enough to support their conclusions, and the paper should address the following issues. I do not recommend to publish it.

1. MALDI MS and gel electrophoresis should be performed to verify the zymogen exchange reaction, not only by luciferase read-out. More direct evidences should be provided to validate the occurrence of the “chain transfer” event wherein part of the LA PDS is transferred from one protein to another.
2. If the protein activator could induce the self-immolation of poly(lipoic acid) to liberate papain from its zymogen, poly(lipoic acid) will be completely degraded into lipoic acid monomers, and protein activator cannot transition into corresponding ZLA through conjugation with LA PDS.
3. Why cannot papain initiate the self-immolation of poly(lipoic acid) during the reaction between papain and LA PDS?
4. The authors should perform more characterizations to validate the occurrence of “chain transfer” reaction between papain and LA PDS.
5. MALDI MS analysis of three bromelain zymogen should be provided in Figure 3A.
6. Gel electrophoresis and SEC-MALS characterization of ZO, ZPEG and ZLA through cryo-polymerization method prepared with papain and bromelain should be provided.
7. More types of characterizations should be provided to prove the efficacy of protein activator, not only by fluorescence read-out.
8. How to stop the zymogen exchange reaction between two proteins? Will the poly(lipoic acid) be completely depolymerized during the exchange reaction?
9. More biological experiments should be performed to verify the application potential of the concept.

REVIEWER COMMENTS

Reviewer #1 (Remarks to the Author):

Comment / question 1 (Q1). This manuscript developed the chemical zymogen (inactive proteins) based on modification of the cysteine moiety of the active protein with sulfhydryl polymers. In the presence of DTT, the polymer chains were degraded into small monomers by a "self-elimination" process, resulting in the full reactivation of the zymogen. In the preparation of zymogen, the polymer was connected to the protein surface via a chain transfer-like pathway. This "chain transfer" pathway was further used to explain the protein-protein activation process. The chains on the zymogen surface were partially transferred to the activator surfaces, and consequently, the spatial site-blocking effect of the polymer chains that exist in the zymogen disappeared with the recovery of protein activity. Conversely, the increased chain on the surface of the activator led to the inhibition of its own activity to achieve zymogen exchange. Despite the interesting finding of this phenomenon, the experimental analysis is insufficient and lacks in-depth exploration, which contributes weakly to the concepts proposed in the article.

Response 1 (R1): we thank the reviewer for an attentive reading of the manuscript. It is pleasing to see that this reviewer recognizes the strength of the proposed chemistry and highlights its "easy-to-use, strong universality, and high stability and efficiency" (see below).

We have conducted an in-depth revision and included a significant amount of new data (3 new composite figures), specifically addressing the questions brought forward by this Reviewer. We hope that our revision fulfills sufficiently to the depth of exploration of the proposed concept.

The followings are the specific concerns.

Q2. The authors demonstrated the concept of zymogen activation by kinetic measurements and point luminescence imaging. The chain transfer occurs between the activated protein and the zymogen, which is essentially an exchange of sulfhydryl groups, thus causing a transition between protein activities. For this reaction, results in the article demonstrate only the existence of this phenomenon without providing any quantitative data.

R2: We thank the reviewer for bringing up these questions and are happy to provide quantitative answers in the revised version of the manuscript.

Q3: what is the molecular weight of the chain required for the reaction to occur;

R3: **New Figure 7** addresses this question quantitatively:

We use MALDI and illustrate that a chain with as few as 2 lipoic acid units sustains the zymogen exchange reaction.

Q4: what is the length change of the chain after activation occurs;

R4: **New Figure 6** addresses this question quantitatively:

We used MALDI to investigate the zymogen exchange reaction and observed that protein activator most typically accepts a sequence of 1-3 lipoic acid monomer units (plus the terminal group). This observation was consistent for at least three different protein activators.

Q5: whether or not various proteins are different for chain lengths and numbers;

R5 : **New figures 6 and 7** illustrate that quantitative observations are consistent between different protein activators (creatine kinase, pyruvate kinase type VII, transglutaminase)

Q6: whether or not a generalized reaction and modification condition can be defined

R6: The answer is YES.

Throughout the experiments, the same generalized conditions were applied to all zymogen exchange reactions;

Q7: the rate and kinetics of the chain transfer reaction should be systemically explored

R7: New Figure 5D addresses this question.

We used LA PDS polymer with thiopyridine terminal groups; this makes chain transfer reaction traceable by UV-vis spectroscopy (monitoring the release of the terminal group, which is the side product of the chain transfer reaction). Kinetics of the process proved to be rather fast and under the chosen experimental conditions the conjugation reached full conversion within 5 minutes (initial rate of reaction: over $10 \mu\text{M}\cdot\text{min}^{-1}$).

Q8. The authors should provide one or more useful applications for the discovered "zymogen exchange" phenomenon.

R8: We thank the reviewer for this comment. We admit that our research was driven by fundamental goals, we aimed to investigate fundamental aspects of the design of chemical zymogens and the zymogen exchange reactions. We did not pursue the development of this platform into a practical application, to work in a specified complicated biological environment.

However, several applications can be proposed:

- Our results demonstrate that efficacy and potency of the activator proteins does not correlate with the computed values of thiol surface accessibility or pK_a . These are new data added to the revised manuscript (Figure 4). Zymogen exchange reaction therefore serves to address a practically important challenge of *experimentally* mapping the reactivity of protein thiols. Thiol mapping is extremely important in e.g. medicinal chemistry, for covalent modifiers used as drugs (See Ref. 39 in the revised manuscript).
- The most important envisioned application is in fundamental science, specifically in the realm of artificial biology. It is a rapidly developing discipline which aims to reconstruct nature using tools of synthetic biology and chemistry (See recent Nature Communication manuscripts 1652 (2020), 6897 (2021)). Zymogen chemistry presented herein is pivotal for the designs of artificial cells, specifically to engineer artificial enzymatic signalling pathways.

We added this information to the revised manuscript (in Conclusions section).

Q9 Also, the specificity of the zymogen activation in the real biological environment is questionable, because sulfhydryl groups widely exist in various bio-species. Therefore, under the complicated conditions of the biological environment, it is necessary to perform independent verification to illustrate its feasibility and significance.

R9. Thiols are indeed abundant in nature, but disulfide exchange most typically requires micromolar content of thiols. In practice, disulfides are stable in serum and are readily degraded only intracellularly, in the presence of a high content of glutathione. Indeed, lipoic acid polydisulfides have been used extensively in biomedicine, predominantly for intracellular protein delivery, including in vivo applications (see Refs. 35 in the revised manuscript). This illustrates that disulfides and LA PDS in particular are stable in "real biological environment".

Most importantly, we highlight that intracellular or in vivo applications were not proposed in our work.

[redacted]

Q10. The authors also proposed that the polymer LA PDS undergoes "chain transfer" and "self-elimination". Therefore, is it possible that the activated protein can still maintain biological activity when its concentration is sufficiently highly or when it possesses exceptionally high amount of sulfhydryl groups, thus achieving real protein "green activation".

R10. We are not sure we understand the question.

Enzyme activity will be observed if the essential thiol is free;
In a multi-thiol protein, “auto-activation” cannot be ruled out.

Q11. The authors seem not to highlight the advantages of protein modification method in their research. In my opinion, the modification process needs to be highlighted as easy-to-use, strong universality, and high stability and efficiency. For highlighting its unique role, it can be compared with common protein modification methods.

R10: We fully agree and thank the reviewer for an invitation to further highlight the advantages of the designed platform. We agree with the assessment that the proposed modification is easy-to-use, universal, and efficient. Brief discussion on pages 6-8 highlights to this end.

Q11. In order to prove the authors' conjecture, it is suggested that the interaction between polymer chains and protein sulfhydryl groups in water be experimentally verified by other theoretical calculations, such as computer simulations to further simulate the structure and "chain transfer" behaviour of the polymer.

R11: To address this question, we have extended the experimental validation of chain transfer to

- MALDI experiments in solutions of two proteins and in mixtures of three proteins (**NEW Figures 6, 7**);
- UV-vis and HPLC monitoring of bioconjugation and activity read-out (**NEW Figure 5**);

and also performed **computer modelling of the proteins** in terms of surface accessibility of thiols, and modelling of LA PDS (partial atomic charges).

The revised manuscript contains a paragraph specifically discussing computer modelling and theoretical considerations for the reactions between LA PDS and proteins (page 8).

We hope that the new material added to the revised manuscript sufficiently characterises the chain transfer reaction.

Reviewer #2 (Remarks to the Author):

The paper ‘Green self-immolative polymer: molecular antenna to collect and propagate the signal for zymogen activation’ investigates the development of chemical zymogens using a series of disulphide coupled additives. Most notably, they use a self-immolative lipoic acid polymer connected through a disulphide bridge. This polymer can undergo depolymerization with a protein activator, thus activating the enzyme. This is a significant advance on previous work as they demonstrate a protein can be used to generate the switch in enzyme activity and thus the science is extremely interesting and topical for a broad audience. However, I had some issues with the clarity of the data presented and thus I believe more detail is needed before their results can be properly assessed. I have highlighted these points below. I believe novelty is high and thus if these points can be addressed it will make this manuscript acceptable for publication in Nature Communications.

Response : we thank the reviewer for the assessment of our submission as one with high novelty, worthy publication in Nature Communication.

Q1. Figure 2B – This data is not clear evidence of the conjugation of the lipoic acid polymer. We would typically see a shift in the SEC trace upon significant increase in molecular weight (like Figure 3C). The one trace shown is in black and it is not explained in the figure, the lines in red over the top (which are supposed to correspond to the starting materials and conjugate) are firstly not SEC traces

and seem on top of each other to me. Please explain in the text how this shows conjugation? They are also hard to tell apart as the colours are very similar.

Q2. During revision, we removed the figure in question as redundant.

We believe that Figure 3 and new Figures 6 and 7 provide quantitative evidence of bioconjugation between LA PDS and proteins.

Q3: Figure 2C&D – The addition of GSH/DTT concentration would add some clarity

R3: Done.

This information is now added to the figure captions (was previously mentioned in the experimental section).

Q4. Figure 2F – This data showed that the activity was not completely blocked for Z, without DTT. This should be discussed briefly in the text.

R4: The Y-axis in this figure (Figure 2B in the revised manuscript) is in log scale; residual enzymatic activity observed in the zymogen formulation is miniscule.

Q5. Figure 3 – The first two enzymes have similar activity when cleaved regardless of the additional moiety. In contrast, creatine kinase has completely different activity for each moiety over the time frame of the experiment, why is this? Does this mean there are differences in cleavage rates?

R5: This is a very insightful observation.

Creatine kinase is a multimeric protein and has several thiols in each protein sub-unit; we believe that blocking multiple thiols leads to the dis-assembly of the kinase tetramer; upon removal of PEG or removal of LA PDS, activity is restored only upon the re-assembly of the dimer. This process takes time, and perhaps is incomplete, thus lowering the overall yield of recovery of activity.

Q4: Figure 4B – The authors state that only the ZLA can be activated by the protein. However, figure 4B appears to show that there is increase in activity for ZPEG as well (approx. half), this doesn't seem to correlate to Figure 4C where I would expect to see a low fluorescence. Can the authors explain this?

R4: We thank the reviewer for an insightful observation. We surmise that this is due to an incomplete blocking of activity within Z(PEG). Nevertheless, recovery of Z(LA) is pronounced far better than Z(PEG), with statistical significance.

Q5: Figure 4 – The authors use pH 8, 25 mM Borate buffer, why is this? It doesn't look like pH is an issue but does high salt effect zymogen switching? I would like to see one experiment done in standard physiological conditions (phosphate buffer pH 7.4) to assess if this approach is useful under these conditions.

R5 : **Done.**

As requested, we have performed zymogen reactivation in PBS.

Zymogen reactivation can be done in PBS, although we do note that the process is somewhat impeded compared to the same reaction in the borate buffer.

From pH 8 borate buffer to pH 7 PBS, changes are in

Reactivation of Z_{LA} and Z₀ in PBS (pH 7.4) using DTT (left) or PK_{VII}.

- pH: disulfide exchange is faster in pH 8;
- pH: the LA coil will be much more open at a higher pH, due to enhanced ionization of the carboxylate;
- saline: the presence of salt decreases the thermodynamic quality of the solvent for hydrophobic polymers (LA PDS) so that the polymer coil will be more collapsed in PBS than borate buffer.

All this taken together means that reactivation can be expected to be lower in PBS than in borate buffer pH 8. This is indeed what we see in the experiment.

Nevertheless, the main conclusions remain:

- zymogens can be reactivated;
- Z_{LA} but not Z_0 are reactivated via the zymogen exchange reaction.

Reviewer 3

The authors prepared three types of zymogens through disulfide linkages including Z0, ZPEG and ZLA. The design strategy was applied to two thiol-containing proteases and a kinase. As a result, the bioactivities of the enzymes were blocked and could be restored by reducing small molecules. Among the zymogens, only ZLA modified by poly(lipoic acid) could be reactivated by protein activators. They also demonstrate a unique zymogen exchange reaction between two proteins. The results and methods reported here have the potential to facilitate the development of new chemical zymogens. However, the data provided here are not enough to support their conclusions, and the paper should address the following issues. I do not recommend to publish it.

Response: we thank the reviewer for recognizing the potential of our findings.

Q1. MALDI MS and gel electrophoresis should be performed to verify the zymogen exchange reaction, not only by luciferase read-out. More direct evidences should be provided to validate the occurrence of the “chain transfer” event wherein part of the LA PDS is transferred from one protein to another.

R1. New figures 6 and 7 address this request.

- We show quantitatively the transfer of LA PDS from one protein to another;
- This phenomenon is further validated by revealing “trans-PEGylation”: transfer of PEG from one protein to another;
- We show that chain transfer can occur over at least 2 consecutive transfer events involving three proteins.

We hope that these two new figures address this request in satisfactory manner.

Q2. If the protein activator could induce the self-immolation of poly(lipoic acid) to liberate papain from its zymogen, poly(lipoic acid) will be completely degraded into lipoic acid monomers, and protein activator cannot transition into corresponding ZLA through conjugation with LA PDS.

R2: New Figure 5.

To address this question quantitatively, we added a new figure in which we investigate the chain transfer mechanism using activity readout and HPLC (to monitor the monomer release).

Reaction between a polymer and LA PDS splits the polymer chain into two, at a random disulfide. Only one part of LA PDS is degraded (as registered via HPLC), whereas the other part is conjugated to the protein.

Q3. Why cannot papain initiate the self-immolation of poly(lipoic acid) during the reaction between papain and LA PDS?

R3: New Figure 5 :

Papain is fully capable of initiating self-immolation of LA PDS.

To prevent this in Figures 6 and 7, we specifically added a protease (thiol) inhibitor.

Q4. The authors should perform more characterizations to validate the occurrence of “chain transfer” reaction between papain and LA PDS.

R4: Done : New Figure 5.

We investigated this reaction in details, some of which are shown in Figure 5.

We investigate the zymogen formation (activity readout); the accompanying release of LA (HPLC); and quantify the rate of reaction via monitoring the end group release.

Q5. MALDI MS analysis of three bromelain zymogen should be provided in Figure 3A.

R5: done

Q6. Gel electrophoresis and SEC-MALS characterization of Z0, ZPEG and ZLA through cryopolymerization method prepared with papain and bromelain should be provided.

R6: Regretfully, papain and bromelain exhibit very strong interaction with the SEC column materials and that in the gel electrophoresis; we were not capable of performing these experiments. We even consulted Wyatt (the world leader in SEC); their technical support staff visited our site but could not resolve this issue. Nevertheless, we believe that MALDI characterization and numerous activity readout data sets in this manuscript provide quantitative proof of the zymogen synthesis and reactivation.

Q7. More types of characterizations should be provided to prove the efficacy of protein activator, not only by fluorescence read-out.

R7: The revised manuscript contains three new figures (5-7) in which we investigate zymogen interconversions using MALDI (molar mass monitoring), UV-vis (end group release during chain transfer), HPLC (accompanying LA release).

Moreover, we included **New Figure 4D** in which papain zymogen reactivation is registered through visualization of papain activity, namely proteolysis on albumin.

We hope that this significant amount of new data addresses this comment in full.

Q8. How to stop the zymogen exchange reaction between two proteins? Will the poly(lipoic acid) be completely depolymerized during the exchange reaction?

R8: Zymogen exchange can be stopped by adding a covalent inhibitor for one of the proteins (as we do for papain in Figures 6 and 7); it can also be stopped by arresting the disulfide exchange through e.g. acidification (c.f. Figure 2E).

Chain transfer event is accompanied by a release of LA (**New Figure 5C**), but depolymerisation of LA PDS is not complete: part of LA PDS is transferred to the acceptor protein, which itself becomes a zymogen (**New Figure 6**).

Q9. More biological experiments should be performed to verify the application potential of the concept.

R9: The revised manuscript contains three **New Figures 5-7** and illustrate such potential applications as chain transfer and trans-PEGylation as well as zymogen exchange in solutions of three proteins. Envisioned applications of these reactions include proximity-based label transfer between proteins (with high relevance for chemical biology) and the design of artificial signal transduction cascades (with the highest relevance for the field of artificial biology and the design of artificial cells).

We hope that the amount of new data added to the manuscript addresses the comments we have received and the manuscript in its significantly revised form becomes publishable in Nature Communications.

REVIEWER COMMENTS

Reviewer #1 (Remarks to the Author):

This article synthesized a novel chemical zymogen via linkage of polydisulfide rich architectures. Through chain transfer mechanism, a new disulfide bond was re-established between the polydisulfide polymer bridged on the surface of the zymogen and the cysteine of the activator protein, which restored the protein activity. This is a fast, accurate, and multi-round process. The authors have addressed most of the issues raised in the initial review and have re-examined the study of chain transfer in more detail. However, before publication could be considered, some additional issues need to be addressed.

1. The order of presentation in the section of Zymogen design and reactivation is a bit confusing. In the second paragraph, the authors characterized the variations of molecular weight and protein activity of three synthesized zymogens (Z0, ZPEG, ZLA) before and after DTT treatment. In the following paragraph, the authors repeated the discussion of the three zymogens and explained the mechanism of protein activity assay for the first time. I suggest to combine these two paragraphs.
2. When using the activated protein for chain transfer, chemical functional groups are ultimately retained at the end of the zymogen (ZL), rendering incomplete recovery of ZL activity (Figure 6B and 4C). Hence, the type of terminal group plays a crucial role in protein activity. It is suggested that the authors compare the various rates of substrates catalyzed by the zymogens with different terminal groups (PEG, iodoacetamide, and methyl) at the end of the chains.
3. To demonstrate the process of dynamic changes in chain transfer, it is suggested to quench the reaction between protein Cys and the polydisulfide using iodoacetamide at different time points. Then, the intercepted substrate and enzyme intermediates can be analyzed by electrospray ionization mass spectrometry (ESI-MS).
4. The molecular weights of the polymers reported in the manuscript are in the range of 20-40 kDa. Does this molecular weight range undergo optimization? Does this also have an effect on the rate of chain exchange between proteins? Please discuss.
5. In protein rate-related Figures, such as 4C, 6B and 3A, it is suggested to add unmodified proteins as a control group.
6. In Figure 3A, the LA PDS group line is almost invisible. The same problem also appears in Figure 5B. Please revise.

Reviewer #2 (Remarks to the Author):

The paper 'Green self-immolative polymer: molecular antenna to collect and propagate the signal for zymogen activation' investigates the development of chemical zymogens using a series of disulphide coupled additives. Most notably, they use a self-immolative lipoic acid polymer connected through a disulphide bridge. This polymer can undergo depolymerization with a protein activator, thus activating the enzyme. They have demonstrated this approach using a broad range of enzymes and I believe it has high novelty and would be of interest for a broad audience. I believe the authors have sufficiently answered my previous comments on this manuscript. I think the data for PBS in response to my comments should be placed in the supporting information as this would be of general interest for people reading this paper. I did have a couple of additional points I would recommend addressing before acceptance.

- The data seems to contain some work where the LA chain on the protein is relatively long (Figure 3B) but very short in other systems (Figure 3A and Figure 6 and 7). I think the authors should include a brief discussion on whether polymer length and characteristics change when using different enzymes. I would suspect they do. This is likely to impact kinetics of the cleavage process (kinetics does seem different across the protein zymogens).

- I understand the data in Figure 7 is very hard to assess clearly as they are very small changes, but I think it is only mildly convincing in its current form. I wonder whether tagging a lipoic acid might be a better way of seeing this change clearly.

Reviewer #3 (Remarks to the Author):

Although the authors only addressed part of the raised issues due to some technical problems, the manuscript further improved and could be published without further revisions.

On behalf of the authors of the manuscript Ms No NCOMMS-21-28953-A
I would like to thank the Reviewers for the second round of revision.

We are happy to see that the manuscript was improved significantly, that we “**have addressed most of the issues**” and/or the manuscript “**could be published without further revision**”.

REVIEWER COMMENTS

Reviewer #1 (Remarks to the Author):

This article synthesized a novel chemical zymogen via linkage of polydisulfide rich architectures. Through chain transfer mechanism, a new disulfide bond was re-established between the polydisulfide polymer bridged on the surface of the zymogen and the cysteine of the activator protein, which restored the protein activity. This is a fast, accurate, and multi-round process. The authors have **addressed most of the issues** raised in the initial review and have re-examined the study of chain transfer in more detail. However, before publication could be considered, some additional issues need to be addressed.

Response: we thank the reviewer for highlighting that we have addressed most of the issues raised during the first round of revision.

Question 1 (Q1). The order of presentation in the section of Zymogen design and reactivation is a bit confusing. In the second paragraph, the authors characterized the variations of molecular weight and protein activity of three synthesized zymogens (Z0, ZPEG, ZLA) before and after DTT treatment. In the following paragraph, the authors repeated the discussion of the three zymogens and explained the mechanism of protein activity assay for the first time. I suggest to combine these two paragraphs.

Response 1 (R1): We thanks the reviewer for this suggestion and we proof-read the manuscript. Paragraph 2 in this section discusses the three types of zymogens as applied to cysteine proteases. Paragraph 3 deals specifically with creatine kinase, and activity assay presented here pertains specifically to the kinase (and is different to that applied in paragraph 2, to the proteases).

Q2. When using the activated protein for chain transfer, chemical functional groups are ultimately retained at the end of the zymogen (ZL), rendering incomplete recovery of ZL activity (Figure 6B and 4C). Hence, the type of terminal group plays a crucial role in protein activity. It is suggested that the authors compare the various rates of substrates catalyzed by the zymogens with different terminal groups (PEG, iodoacetamide, and methyl) at the end of the chains.

R2: We believe results in Figure 2C address this question: we have analyzed Z(LA) in which LA PDS contains diverse terminal groups and observed that in each case, zymogen reactivation is rather similar.

We note that PEG and methyl (in Figure 4C) are not terminal groups for LA PDS but are connected to the Cys thiol via direct disulfide linkage.

Q3. To demonstrate the process of dynamic changes in chain transfer, it is suggested to quench the reaction between protein Cys and the polydisulfide using iodoacetamide at different time points. Then,

the intercepted substrate and enzyme intermediates can be analyzed by electrospray ionization mass spectrometry (ESI-MS).

R3: To our understanding, one cannot expect to intercept enzyme intermediate and only two states of Cys thiol are possible: the pristine thiol, and the resulting zymogen.

Dynamic changes can be expected only for the decomposition of LA PDS (as a result of the chain transfer, see Figure 5A), but this fast process only results in multimers of the polydisulfide. We believe this analysis falls outside the scope of our work on chemical zymogens.

Q4. The molecular weights of the polymers reported in the manuscript are in the range of 20-40 kDa. Does this molecular weight range undergo optimization? Does this also have an effect on the rate of chain exchange between proteins? Please discuss.

R4: Control over LA PDS molar mass has been reported only in solitary examples (most notably in Lu et al, JACS 2020, 142, 1217) and as such remains a challenge. In our hands, LA PDS synthesized in solution had molar mass 20-40 kDa, but when synthesized via grafting-from the protein, LA PDS was significantly shorter (see Figures 3A, 6, 7).

We agree that this characteristic of LA PDS may prove important. However, the data in Figures 1C and 5C illustrate that depolymerization of LA PDS is very fast and from the standpoint of practical applications, this is not a limiting step.

Q5. In protein rate-related Figures, such as 4C, 6B and 3A, it is suggested to add unmodified proteins as a control group.

R5. We fully agree that protein control is highly important; it is featured in the overall majority of figures (where relevant), including the figures mentioned by the reviewer: Figures 2 B,C in which we first present the zymogen concept; Figure 3A contains the protein control in the MALDI sub-figures, and Figures 3B-D, Figure 4D, Figure 6B.

Q6. In Figure 3A, the LA PDS group line is almost invisible. The same problem also appears in Figure 5B. Please revise.

R6. We thank the reviewer for pointing it out; figures were updated to make all lines of the same thickness.

Reviewer #2 (Remarks to the Author):

The paper 'Green self-immolative polymer: molecular antenna to collect and propagate the signal for zymogen activation' investigates the development of chemical zymogens using a series of disulphide coupled additives. Most notably, they use a self-immolative lipoic acid polymer connected through a disulphide bridge. This polymer can undergo depolymerization with a protein activator, thus activating the enzyme. They have demonstrated this approach using a broad range of enzymes and I believe it has high novelty and would be of interest for a broad audience. **I believe the authors have sufficiently answered my previous comments on this manuscript.** I think the data for PBS in response to my comments should be placed in the supporting information as this would be of general interest for

people reading this paper. I did have a couple of additional points I would recommend addressing before acceptance.

Response: we thank the reviewer for highlighting that we have addressed the comments sufficiently, and are happy to include the data on zymogen reactivation in PBS in the supporting information.

Q1 The data seems to contain some work where the LA chain on the protein is relatively long (Figure 3B) but very short in other systems (Figure 3A and Figure 6 and 7). I think the authors should include a brief discussion on whether polymer length and characteristics change when using different enzymes. I would suspect they do. This is likely to impact kinetics of the cleavage process (kinetics does seem different across the protein zymogens).

R1: We thank the reviewer for inviting us to discuss this notion. Indeed, LA PDS degree of polymerization may exhibit a batch-to-batch variation. However, LA PDS is a fast-depolymerizing polymer (Figure 1) and we don't anticipate molar mass of LA PDS to be critically important for kinetics of zymogen reactivation.

The apparent difference in reactivation kinetics for Z(LA) between e.g. papain and creatine kinase may be due to the readout protocols. Indeed, the kinase activity quantification is a coupled, two-enzyme readout. Also, creatine kinase is a multimeric protein and sub-unit assembly is required for activity, which will also affect the apparent kinetics of zymogen reactivation observed in our experiments.

Lastly, we note that Figure 3B (gel electrophoresis) showed a characteristic "smear" for Z(LA), which we attribute to an interaction between LA PDS and the gel material rather than a high molar mass of Z(LA).

Q2: I understand the data in Figure 7 is very hard to assess clearly as they are very small changes, but I think it is only mildly convincing in its current form. I wonder whether tagging a lipolic acid might be a better way of seeing this change clearly.

R2. We appreciate this invitation to revise Figure 7.

For a more convincing illustration of two round of LA PDS chain transfer, MALDI data are now supported by visualization of enzyme activity for creatine kinase (CK), which decreases upon a chain transfer TO this enzyme, and then increases upon chain transfer FROM it. We hope that this adds clarity and degree of convincing.

Reviewer #3 (Remarks to the Author):

Although the authors only addressed part of the raised issues due to some technical problems, the manuscript further improved and **could be published without further revisions**.

REVIEWERS' COMMENTS

Reviewer #1:

[In remarks to the Editorial office, the reviewer stated that the manuscript is ready for publication as-is.]

Reviewer #2 (Remarks to the Author):

The revised manuscript has suitably addressed my questions and I believe it is suitable for publication in its current form. I would note that the revised Figure 7 is convincing for a cascade, however I believe the authors are claiming that even after reaction with PKVII they still have a CK-ZLa (just a smaller LA polymer) so why does the CK activity appear to switch on completely in Figure 7A? I think this should be addressed in the manuscript as it makes it a little unclear.

REVIEWER COMMENTS

Reviewer #1:

[In remarks to the Editorial office, the reviewer stated that the manuscript is ready for publication as-is.]

Response : We thank this reviewer for all the constructive suggestions and are happy to learn that our revisions were fully satisfactory.

Reviewer #2 (Remarks to the Author):

The revised manuscript has suitably addressed my questions and I believe it is suitable for publication in its current form. I would note that the revised Figure 7 is convincing for a cascade, however I believe the authors are claiming that even after reaction with PKVII they still have a CK-Z1a (just a smaller LA polymer) so why does the CK activity appear to switch on completely in Figure 7A? I think this should be addressed in the manuscript as it makes it a little unclear.

Response: We thank the reviewer for all the constructive comments and suggestions for manuscript improvement, and for advising that our work is suitable for publication in its current form.

Regarding Figure 7A : results shown in this panel have been obtained at a low substrate-to-enzyme ratio; similar levels of luminescence (meaning similar content of the de novo synthesized ATP) in these end-point measurements imply a full ADP-to-ATP conversion, which can be achieved even if the enzyme concentrations in the wells are not identical.